# Chromatin-bound cGAS is an inhibitor of DNA repair and hence accelerates genome destabilization and cell death

Hui Jiang[1] (ID), Xiaoyu Xue[2,3], Swarupa Panda[1] (ID), Ajinkya Kawale[2], Richard M Hooy[4], Fengshan Liang[2], Jungsan Sohn[4], Patrick Sung[2,5] & Nelson O Gekara[1,6,*] (ID)

## Abstract

DNA repair via homologous recombination (HR) is indispensable for genome integrity and cell survival but if unrestrained can result in undesired chromosomal rearrangements. The regulatory mechanisms of HR are not fully understood. Cyclic GMP-AMP synthase (cGAS) is best known as a cytosolic innate immune sensor critical for the outcome of infections, inflammatory diseases, and cancer. Here, we report that cGAS is primarily a chromatin-bound protein that inhibits DNA repair by HR, thereby accelerating genome destabilization, micronucleus generation, and cell death under conditions of genomic stress. This function is independent of the canonical STING-dependent innate immune activation and is physiologically relevant for irradiation-induced depletion of bone marrow cells in mice. Mechanistically, we demonstrate that inhibition of HR repair by cGAS is linked to its ability to self-oligomerize, causing compaction of bound template dsDNA into a higher-ordered state less amenable to strand invasion by RAD51-coated ssDNA filaments. This previously unknown role of cGAS has implications for understanding its involvement in genome instability-associated disorders including cancer.

**Keywords** cancer; cell death; cGAS; chromatin compaction; DNA repair
**Subject Categories** DNA Replication, Repair & Recombination; Immunology
**The EMBO Journal (2019) 38: e102718**

## Introduction

Both the innate immune system—the inherent ability to rapidly sense and respond to infections—and the DNA damage response (DDR), which reacts to threats to our genome, function as surveillance systems essential for preserving the integrity of organisms.

Emerging evidence indicates that these two systems are interdependent (Hartlova *et al*, 2015) and that defects in these two systems lie at the heart of many diseases such as infections, autoimmunity, cancer, and aging-associated disorders including neurodegeneration (Hartlova *et al*, 2015; Erttmann *et al*, 2016). The molecular factors involved in the cross-talk between the DDR and the innate immune system, as well as the underlying mechanisms, remain poorly defined.

The cyclic GMP-AMP synthase cGAS is well known as innate immune sensor that surveys the cytosol for the presence of microbial DNA (Gao *et al*, 2013b; Sun *et al*, 2013), as well as self-DNA released from the nucleus under conditions of genomic stress (Hartlova *et al*, 2015) or from stressed mitochondria (West *et al*, 2015). Upon recognition of double-stranded DNA (dsDNA), cGAS catalyzes the cyclization of ATP and GTP into the second messenger cyclic GMP–AMP (2′3′-cGAMP) (Ablasser *et al*, 2013; Civril *et al*, 2013; Diner *et al*, 2013; Gao *et al*, 2013a,b; Li *et al*, 2013; Sun *et al*, 2013; Zhang *et al*, 2013). Subsequently, cGAMP binds to its adaptor STING (Stimulator of Interferon Genes) (Ishikawa & Barber, 2008), leading to the activation of downstream innate immune responses. Dysfunctions in the cGAS-STING pathway have been implicated in many disorders including infections, inflammatory diseases, neurodegeneration, and cancer (Barber, 2015; Chen *et al*, 2016).

cGAS is generally considered as a cytosolic protein but transiently accumulates in the nucleus following mitotic nuclear membrane dissolution (Yang *et al*, 2017; Gentili *et al*, 2019; Zierhut *et al*, 2019). Moreover, cGAS has also been reported to actively translocate from the cytosol into the nucleus upon DNA damage (Liu *et al*, 2018) but also localizes to the plasma membrane in some cell types (Barnett *et al*, 2019). In spite of these reports, the subcellular localization and function of cGAS in different biological conditions remain hotly debated (Gekara & Jiang, 2019).

Double-strand DNA breaks (DSB) are potentially highly deleterious lesions. If improperly repaired, DSB results in chromosomal deletions or translocations culminating in genome instability-

1   The Laboratory for Molecular Infection Medicine Sweden (MIMS), Umeå Centre for Microbial Research (UCMR), Umeå University, Umeå, Sweden
2   Department of Molecular Biophysics and Biochemistry, Yale University School of Medicine, New Haven, CT, USA
3   Department of Chemistry and Biochemistry, Texas State University, San Marcos, TX, USA
4   Department of Biophysics and Biophysical Chemistry, Johns Hopkins University School of Medicine, Baltimore, MD, USA
5   Department of Biochemistry and Structural Biology, University of Texas Health Science Center at San Antonio, San Antonio, TX, USA
6   Department of Molecular Biosciences, The Wenner-Gren Institute, Stockholm University, Stockholm, Sweden
    *Corresponding author. Tel: +46 729430478; E-mails: nelson.gekara@mims.umu.se; nelson.gekara@su.se

associated disorders including tumorigenesis, accelerated aging, and other diseases (Jackson & Bartek, 2009). DSB repair occurs via two major pathways: non-homologous end-joining (NHEJ) and homologous recombination (HR) (Chapman *et al*, 2012; Ceccaldi *et al*, 2016). NHEJ is an error-prone repair pathway active throughout the cell cycle, and it entails the ligation of DNA ends and often leads to deletion or insertional mutations (Chapman *et al*, 2012; Ceccaldi *et al*, 2016). On the other hand, HR is an accurate repair process active in proliferating cells and occurs mainly during the S and G2 cell cycle phases wherein it engages the undamaged sister chromatid to template break repair to restore the original DNA sequence (Chapman *et al*, 2012; Ceccaldi *et al*, 2016). For a healthy outcome, activation of these pathways is carefully calibrated to ensure timely removal of damaged DNA breaks, but, if the DNA damage is excessive, to promote the induction of cell death to eradicate genetically altered cells (Vitale *et al*, 2011). The regulatory molecules involved in this delicate balance are not fully known. In this study, we identify a new regulator of DNA repair: cGAS. We demonstrate that cGAS is constantly present in the nucleus as a chromatin-bound protein where it acts as a negative regulator of homologous recombination-mediated DNA repair, thereby accelerating micronucleus generation and death of cells under genomic stress. We show that this non-canonical cGAS function is independent of the STING axis.

## Results

### cGAS is constitutively present in the nucleus

While monitoring the subcellular localization of endogenous cGAS in bone morrow-differentiating monocytes (BMDMos) (Fig 1A–I) or exogenously expressed GFP-tagged human cGAS (GFP-hcGAS) in HEK293 cells (Fig EV1A–C), we noticed that cGAS is primarily in the nucleus. Because transient accumulation of cGAS in the nucleus following mitotic nuclear membrane dissolution has previously been observed (Yang *et al*, 2017; Gentili *et al*, 2019), we asked whether nuclear localization of cGAS was sensitive to changes in cell cycle. For that, we enriched cells at G0/G1 cell cycle phase by culturing them at high cell density to induce cell contact inhibition (Figs 1A, D and G, and EV1A), or in serum-free medium (Figs 1B, E and H, and EV1B), or arrested them at G1/early S phase using the DNA polymerase α inhibitor aphidicolin (Figs 1C, F and I, and EV1C). Although showing a slightly increased presence in the cytosol in cells arrested at G0/G1 or G1/early S boundary, cGAS was still abundant in the nucleus (Figs 1 and EV1). This demonstrates that cGAS is constitutively present in the nucleus and cytosol and that the relative abundance of cGAS in these subcellular compartments can to some extent be affected by the cell cycle. To verify the nuclear localization of cGAS, we examined additional cell types including the human THP1 monocytes and the HeLa cells, mouse Raw 264.7 macrophages, and bone marrow-derived macrophages (BMDMs). We found cGAS to be abundant in both the nucleus and cytosol of these different cell types (Fig EV1D).

To elucidate the cGAS features essential for its nuclear localization, we analyzed different cGAS mutants. The nuclear localization of the catalytically dead E225A/D227A mutant (GFP-hcGAS$^{\Delta cGAMP}$) (Raab *et al*, 2016) and the oligomerization-defective K394E mutant (GFP-hcGAS$^{\Delta Oligo}$) (Li *et al*, 2013) was comparable to that of the

wild-type form GFP-hcGAS. In contrast, the DNA binding C396A/C397A mutant (GFP-hcGAS$^{\Delta DNA}$) (Ablasser *et al*, 2013; Kranzusch *et al*, 2013) showed a decreased nuclear presence and was unaffected by the cell cycle phase (Fig EV2A and B). These findings demonstrate that association of cGAS with genomic DNA is required for its nuclear localization. To interrogate this further, we asked whether introduction of a strong nuclear export or import signal (NES or NLS, respectively) would impact cGAS localization. The NLS localized cGAS almost entirely in the nucleus, whereas the NES substantially increased the cytosolic localization of cGAS but did not eliminate its presence in the nucleus (Fig EV2C and D). Taken together with the behavior of the cGAS$^{\Delta DNA}$ mutant (Fig EV2A), these results confirm that the localization and retention of cGAS in the nucleus is due to its association with genomic DNA and, as such, even a strong NES is insufficient to exclude cGAS from the nucleus.

### Nuclear cGAS accelerates genome destabilization, micronucleus generation, and cell death

We wished to determine the biological role of nuclear cGAS. Micronuclei are a hallmark of genome instability. Micronuclei arise following the mis-segregation of broken chromosomes during mitosis (Crasta *et al*, 2012; Gekara, 2017; Mackenzie *et al*, 2017) and have recently been described as platforms for cGAS-mediated innate immune activation following DNA damage (Bartsch *et al*, 2017; Gekara, 2017; Harding *et al*, 2017; Mackenzie *et al*, 2017). We found that in response to γ-irradiation, HEK293 cells expressing GFP-hcGAS exhibit a higher incidence of micronuclei than cells expressing a GFP control containing a nuclear localization sequence (GFP-NLS) (Fig 2A and B). As expected, GFP-hcGAS, but not GFP-NLS, restored *IFNB1* response to transfected DNA (Fig 2C). This observation led us to hypothesize that the presence of cGAS in the nucleus and micronucleus generation was causally related. Hence, we tested whether endogenous cGAS promotes micronucleus generation in bone marrow-differentiating monocytes (BMDMos). To induce micronucleus generation, BMDMos were synchronized in G2/M phase using the microtubule-depolymerizing agent nocodazole followed by γ-irradiation then released (Fig 2D). BMDMos from WT mice exhibited more micronuclei compared to those from *cGAS*$^{-/-}$ mice (Fig 2E and F), demonstrating a role for cGAS in accelerating genomic destabilization and micronucleus generation. Noteworthy, we observed in addition to the nucleus all the micronuclei were cGAS-positive. Based on these data, we conclude that micronucleus-associated cGAS emanates mainly from the nucleus as opposed to an influx from the cytosol post-micronucleus membrane rapture (Harding *et al*, 2017; Mackenzie *et al*, 2017).

To further elucidate the biological relevance of nuclear cGAS, we tested the impact of cGAS on DNA damage-induced cell death. BMDMos from *cGAS*$^{-/-}$ mice were resistant to irradiation-induced cell death compared to those from WT mice (Fig 2G). Thus, by accelerating genomic destabilization, micronucleus generation, and cell death, cGAS likely contributes to the elimination of cells with severely damaged genomes.

### cGAS impedes DNA repair independently of STING

Next, we sought to determine whether cGAS contributes to genomic instability by inhibiting DNA double-strand break (DSB)

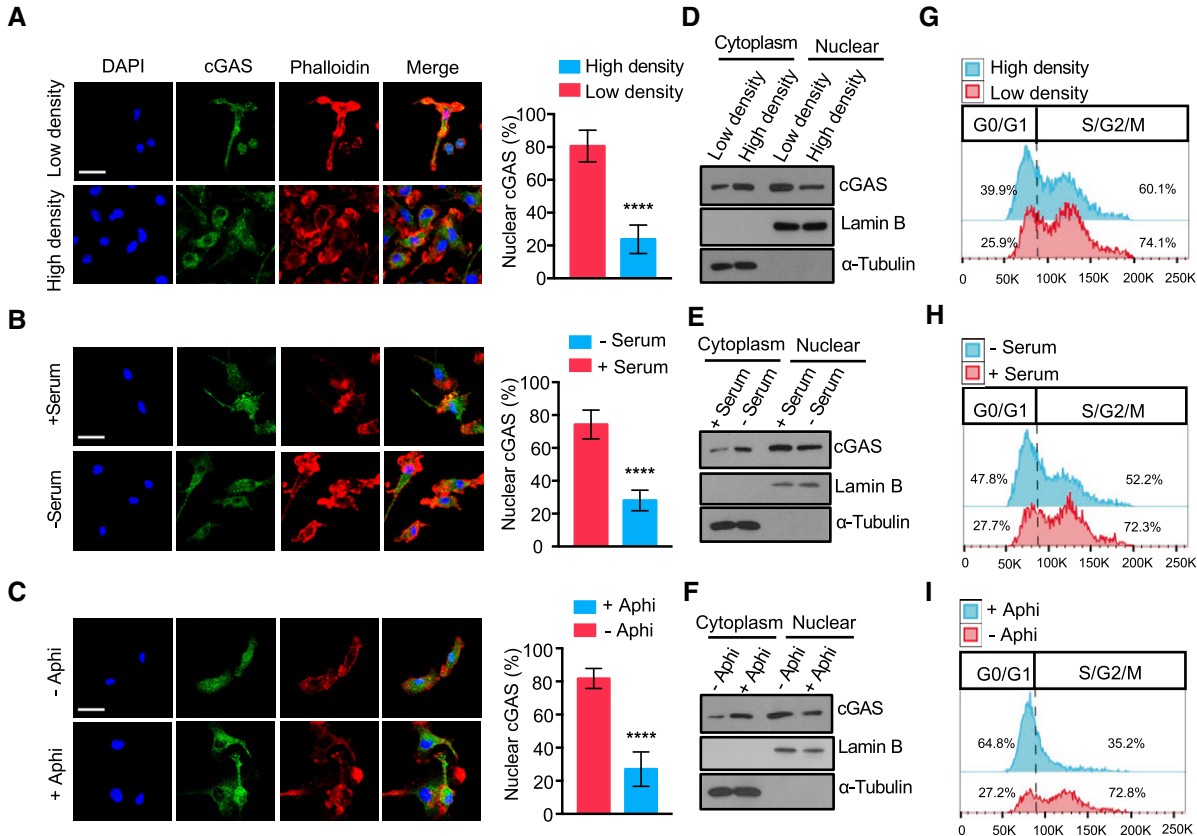

**Figure 1. cGAS is constantly present in the cytosol and nucleus and is impacted by cell cycle.**

A–C   Left: Immunofluorescence images of cGAS in the nucleus (DAPI) and cytosol (phalloidin) in BMDMos cultured at low/high density (A), with/without serum (B), or with/without aphidicolin (Aphi) (C). Scale bar: 50 μm. Right: Corresponding quantification of the nuclear cGAS from 6 different fields with *n* > 50 cells.

D–F   Immunoblot estimation of cGAS in nuclear/cytosolic subcellular fraction of BMDMos cultured under indicated conditions. Lamin B and α-tubulin are nuclear and cytosolic markers, respectively.

G–I   Flow cytometric analysis of cell cycle of BMDMos depicted in (D–F).

Data information: Data are presented as means ± SEM. Statistical significance was assessed using unpaired Student's *t*-test. ****P ≤ 0.0001.
Source data are available online for this figure.

repair and whether this was via the canonical STING pathway. For that, we monitored the levels of DSBs at different time points following γ-irradiation by comet and pulsed-field gel electrophoresis assays. We found that γ-irradiated BMDMos from *cGAS*⁻/⁻ mice are more adept at resolving DSBs than those from WT mice (Figs 3A and B and EV3A). Curiously, while γ-irradiated BMDMos from *cGAS*⁻/⁻ mice exhibited faster DSB resolution, those from *Sting*⁻/⁻ mice were comparable to WT BMDMos in this regard (Fig 3A and B). Furthermore, expression of GFP-hcGAS in HEK293T cells that lack endogenous cGAS and STING (Sun *et al*, 2013) impaired DSB repair in these cells (Fig EV3B), but, as expected, failed to restore the *IFNB1* response (Fig EV3C). Thus, while essential for the induction of inflammatory genes following DNA damage via STING (Hartlova *et al*, 2015; Erdal *et al*, 2017), cGAS also promotes DNA damage by inhibiting DSB repair independently of STING.

To further interrogate how cGAS affects genome stability, we considered whether inhibition of DSB repair was mediated by cGAMP via a hitherto undefined STING-independent mechanism. However, treatment of HEK293 cells with cGAMP before or during γ-irradiation did not lead to increased fragmentation of genomic DNA (Fig EV3D), but, as expected, activated STING-dependent interferon regulatory factor (IRF3) (Fig EV3E). Accordingly, the full-length and catalytically dead GFP-hcGAS^ΔcGAMP comparably inhibited DSB repair in HEK293 cells, as assessed by comet tail length (Fig EV3F and G). As expected, unlike GFP-hcGAS, GFP-hcGAS^ΔcGAMP failed to restore the IFN-I response (Fig EV3H) but boosted micronucleus generation (Fig EV3I and J). Further, analysis of WT, *cGAS*⁻/⁻, *Sting*⁻/⁻, and *cGAS*⁻/⁻ *Sting*⁻/⁻ BMDMos revealed that cGAS-driven micronucleus generation and cell death are independent of STING (Fig 3C–F). Finally, when we compared human cGAS (hcGAS) and mouse cGAS (mcGAS), we found them to inhibit DNA repair comparably when exogenously expressed in HEK293 cells (Fig EV3K).

## cGAS drives γ-irradiation-induced bone marrow ablation independently of STING

To establish the *in vivo* physiological relevance of cGAS-mediated inhibition of DNA repair, we examined the depletion of bone

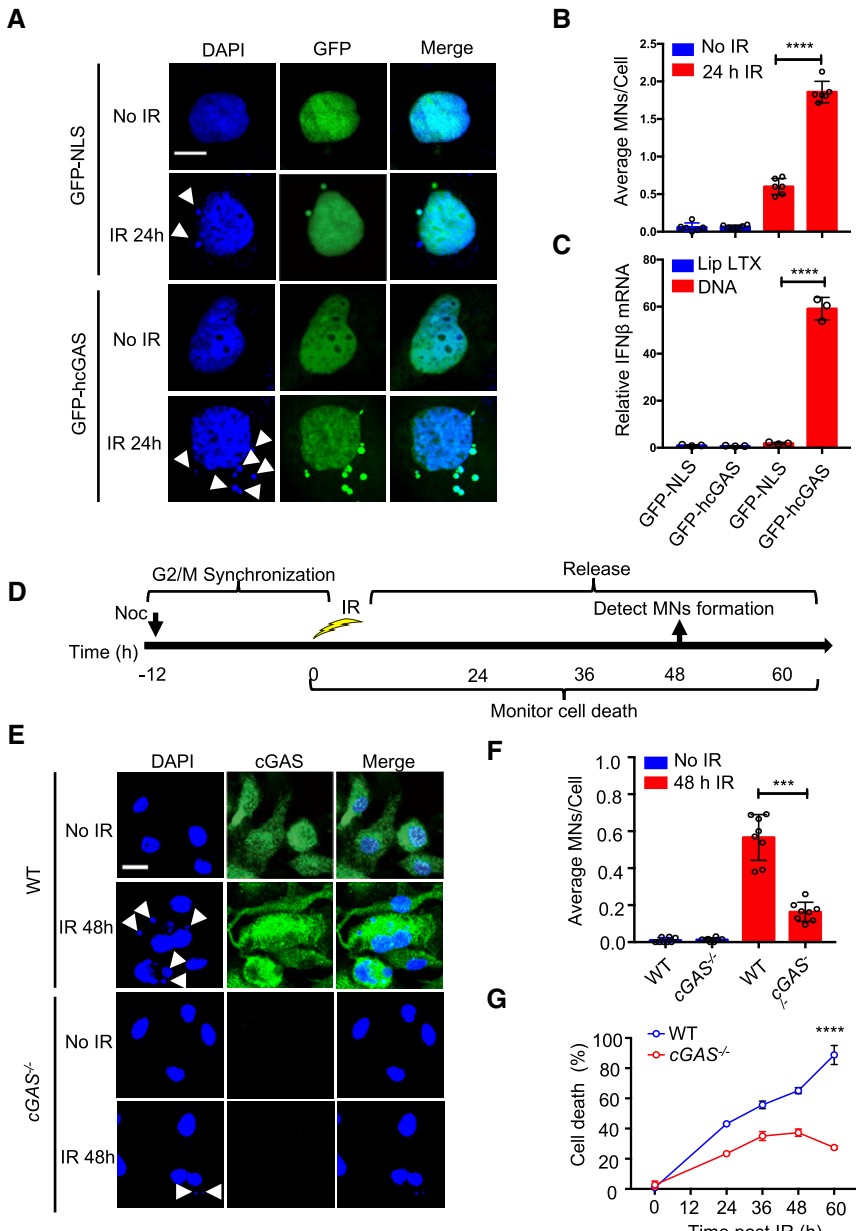

**Figure 2. cGAS promotes irradiation-induced micronucleus generation and cell death.**

A, B    Micronuclei (indicated by arrowhead) in GFP-NLS- or GFP-hcGAS-expressing HEK293 cells before (0 h) or 24 h after γ-irradiation (IR; 10 Gy). Scale bar: 10 μm (A).
        (B) The average MNs/cell. Graphs show mean ± SEM (*n* = 3 independent experiments) representing six different microscopic fields with over 200 cells.
C        IFNB1 response in HEK293 cells stimulated with transfected plasmid DNA. Mean ± SEM of *n* = 3 independent experiments.
D        Experimental outline for micronucleus generation and cell death after γ-irradiation.
E        Micronucleus (indicated by arrowhead) and cGAS staining in WT and cGAS$^{-/-}$ BMDMos exposed to γ-irradiation (10 Gy). Scale bar: 10 μm.
F        Average MNs/cell in BMDMos. MN graphs show mean ± SEM (*n* = 3 independent experiments) representing eight different microscopic fields with over 200 cells.
G        Cell death in WT and cGAS$^{-/-}$ BMDMos that were first synchronized at G2/M, then γ-irradiated (10 Gy) followed by release and analysis at indicated time points.
        Mean ± SD, x biological triplicates (*n* = 3) per treatment group are shown.

Data information: Statistical significance in (B), (C), and (F) was assessed using unpaired two-tailed Student's *t*-test. ***$P \leq 0.001$ and ****$P \leq 0.0001$. Statistical significance in (G) was assessed using two-way ANOVA test, ****$P < 0.0001$.
Source data are available online for this figure.

marrow cells in mice following γ-irradiation. First, by analyzing wild-type mice, we found that following acute γ-irradiation (9 Gy), over 90% of bone marrow cells are depleted within the

first 36 h (Fig 4A–D). When WT and *cGAS*$^{-/-}$ mice were compared 10 h after γ-irradiation, *cGAS*$^{-/-}$ mice were found to be more resistant to γ-irradiation-induced bone marrow cell

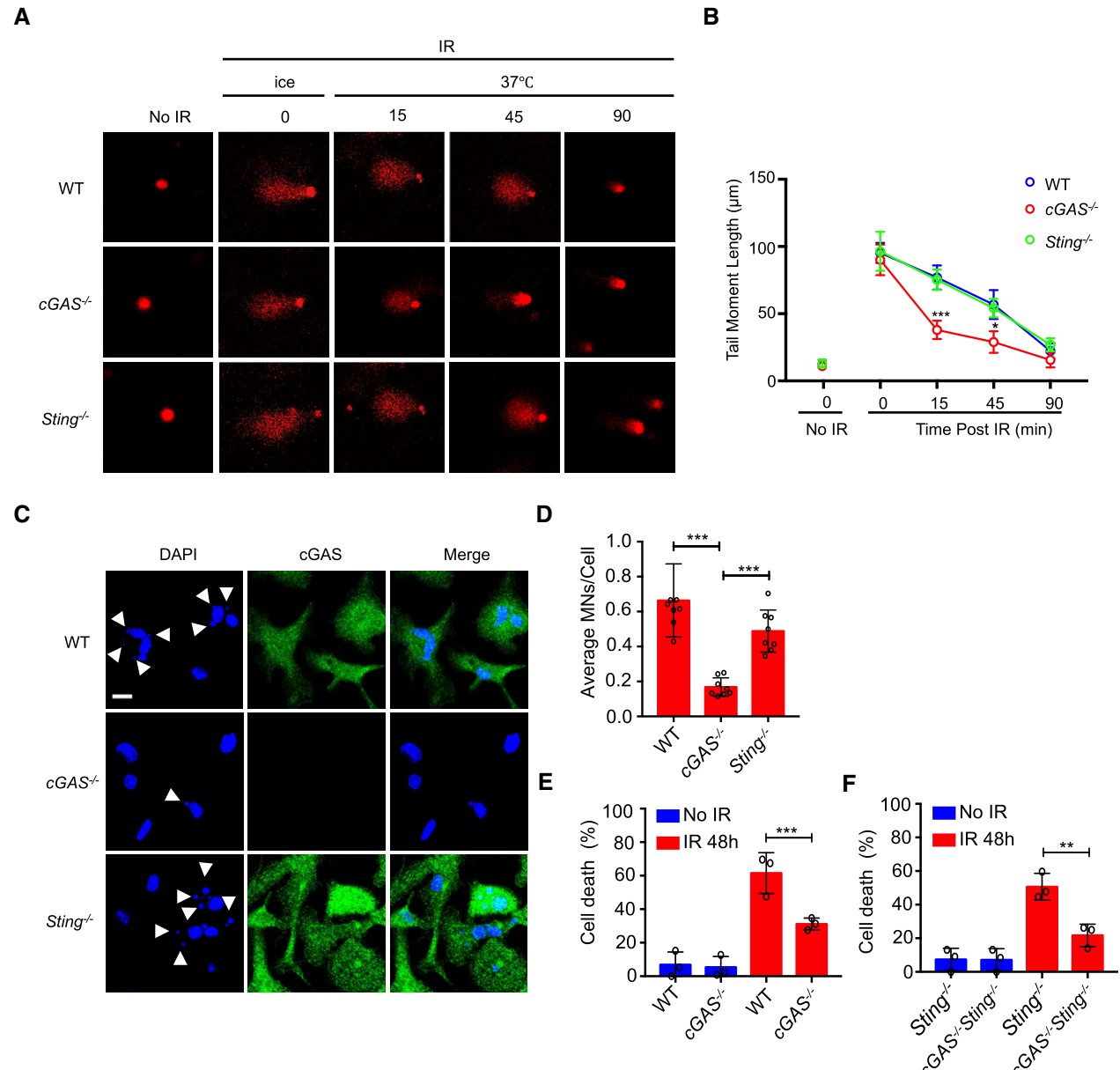

**Figure 3. cGAS inhibits DNA damage repair and promotes micronucleus generation and cell death independently of STING.**

A, B  BMDMos from cGAS$^{-/-}$ mice exhibit enhanced DNA repair efficiency than those from WT and Sting$^{-/-}$ mice. (A) Representative comet tails of WT, cGAS$^{-/-}$, and Sting$^{-/-}$BMDMos exposed to γ-irradiation (IR: 10 Gy) on ice, then incubated at 37°C for indicated duration. (B) Corresponding quantification of the comet tail moments from 20 different fields with $n > 200$ comets of three independent experiments.

C, D  cGAS promotes micronucleus generation in BMDMos independently of STING. (C) Confocal microscopic visualization of micronucleus (indicated by arrowhead) and cGAS staining in WT, cGAS$^{-/-}$, and Sting$^{-/-}$ BMDMos exposed to γ-irradiation (10 Gy). Scale bar: 10 μm. (D) Average MNs/cell in corresponding representative images. Bar graphs show mean values from eight different microscopic fields with over 200 cells.

E, F  γ-Irradiation-induced cell death in WT and cGAS$^{-/-}$ BMDMos (E). γ-Irradiation-induced cell death in Sting$^{-/-}$ and cGAS$^{-/-}$ Sting$^{-/-}$ BMDMos (F).

Data information: Data are presented as mean ± SEM of $n = 3$ independent experiments. Statistical significance was assessed using two-way ANOVA in (B) and one-way ANOVA in (D), (E), and (F) followed by Sidak's post-test. *$P < 0.05$, **$P < 0.01$, ***$P < 0.001$, and ****$P < 0.0001$.

Source data are available online for this figure.

depletion (Fig 4E–H). Further, compared to Sting$^{-/-}$ mice, Sting$^{-/-}$ cGAS$^{-/-}$ mice were more resistant to γ-irradiation-induced bone marrow ablation (Fig 4I–L), confirming that cGAS-mediated inhibition of DNA repair and accelerated cell death *in vivo* is independent of the STING pathway.

**cGAS attenuates HR-DNA repair via DNA binding and self-oligomerization**

DSB repair occurs via two major pathways: non-homologous end-joining (NHEJ) that is active throughout the cell cycle and

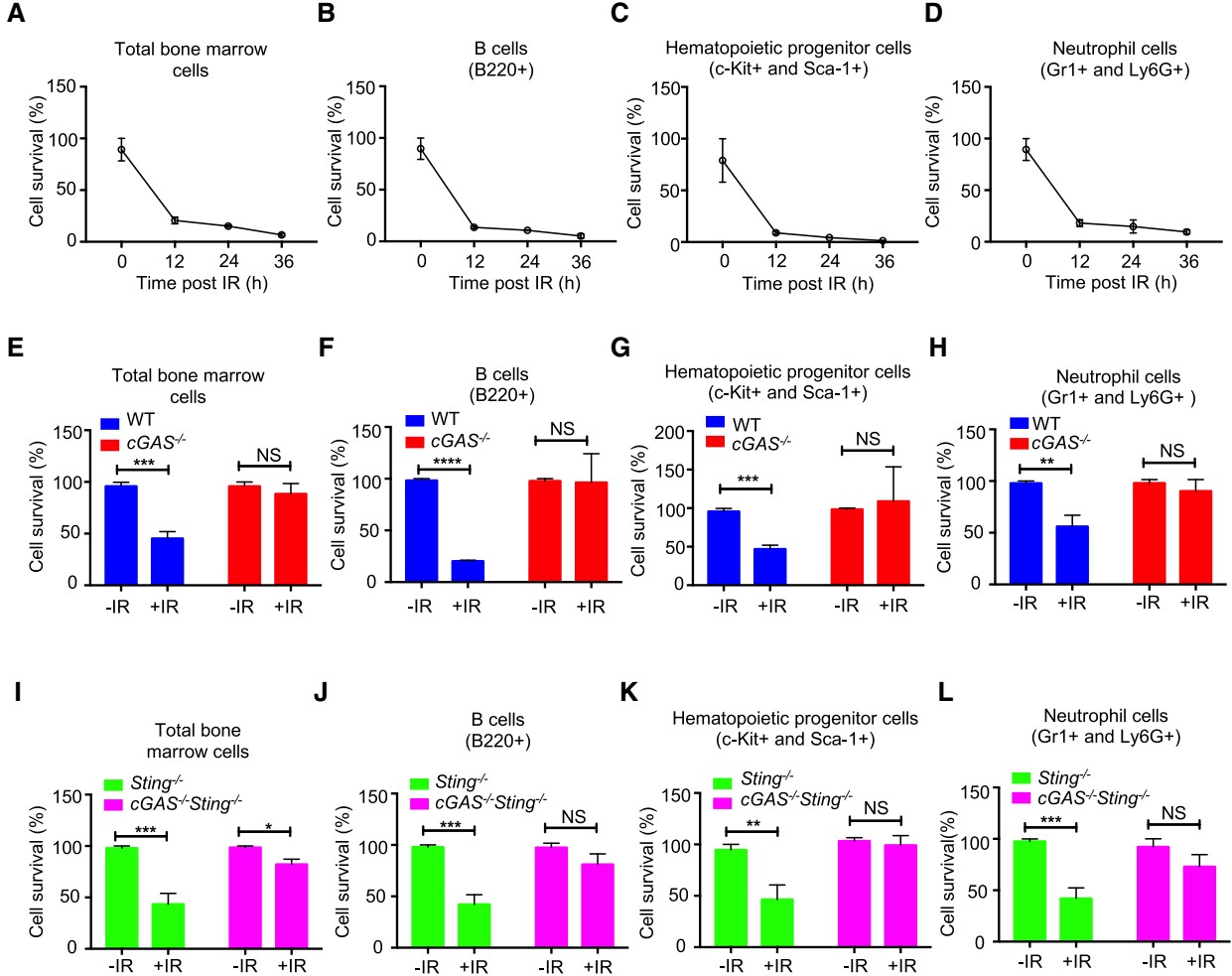

**Figure 4. cGAS accelerates γ-irradiation-induced depletion of bone marrow cells independently of STING.**

A–D  Kinetics of *in vivo* depletion of indicated bone marrow cells in WT mice (*n* = 3) after γ-irradiation (9 Gy).
E–H  Bone marrow cells in WT (*n* = 3) and cGAS⁻/⁻ (*n* = 3) mice 10 h post-γ-irradiation.
I–L  Bone marrow cells in Sting⁻/⁻ (*n* = 3) and cGAS⁻/⁻ Sting⁻/⁻ (*n* = 3) mice 10 h post-γ-irradiation.

Data information: Data in this figure are presented as mean ± SD. One-way ANOVA followed by Sidak's post-test. NS: *P* > 0.05, *\*P* ≤ 0.05, *\*\*P* ≤ 0.01, *\*\*\*P* ≤ 0.001, and *\*\*\*\*P* ≤ 0.0001.

homologous recombination (HR) that occurs during the S and G2 cell cycle (Chapman *et al*, 2012; Ceccaldi *et al*, 2016). To determine which of these repair pathways is impeded by cGAS, firstly we asked whether cGAS impacts cell cycle. Cell cycle analysis revealed that WT and *cGAS⁻/⁻* BMDMos were comparable and that up to 81% of cells were in the HR-competent S/G2 phase (Appendix Fig S1A). Interestingly, arresting cells at the G1/early S phase using aphidicolin abolished the inhibitory effect of cGAS on DSB repair (Appendix Fig S1B and C), indicating that cGAS was likely targeting the HR pathways and that this inhibition was not due to difference in cell cycle. To specifically evaluate the DSB repair pathway impeded by cGAS, we examined the repair of a site-specific DSB induced by the I-SceI endonuclease using the direct repeat-GFP (DR-GFP) (Pierce *et al*, 2001) and the total-NHEJ-GFP (EJ5-GFP) (Bennardo *et al*, 2008) reporter systems for HR and NHEJ, respectively (Fig 5A and B). siRNA knockdown of endogenous cGAS in U2OS

cells increased HR efficiency but minimally affected NHEJ repair (Fig 5C and D). Histone H1 is a negative regulator of HR-mediated DNA repair. Therefore, as control we also silenced histone H1.2 in U2OS cells. Knockdown of histone H1.2 similarly increased HR repair efficiency (Fig 5C and D). In contrast, exogenous expression of cGAS in HEK293T cells strongly reduced the efficiency of HR but had minimal effect on NHEJ repair (Fig 5E and F).

We tested several cGAS mutants in order to elucidate its functional features required for HR inhibition. The catalytically dead hcGAS^ΔcGAMP inhibited HR to a similar degree as wild-type hcGAS. In contrast, the DNA binding (hcGAS^ΔDNA) and oligomerization (hcGAS^ΔOligo) mutants both lacked such inhibitory effect (Fig 5E and F). These data demonstrate that cGAS specifically blocks HR-mediated DNA repair via its DNA binding and oligomerization but not catalytic activity. To further develop these findings, we addressed the importance of cGAS nuclear

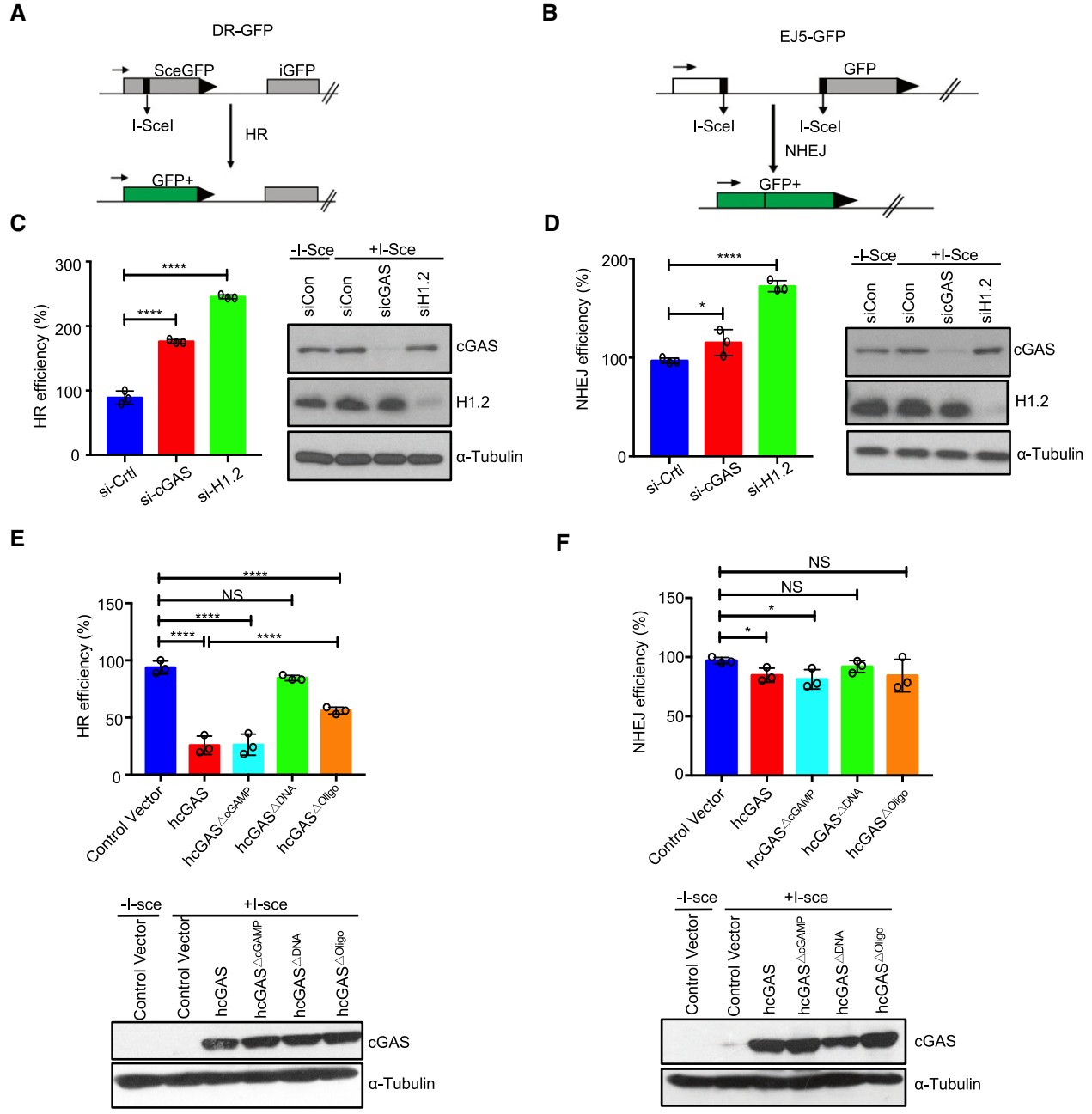

**Figure 5. DNA binding and subsequent oligomerization of cGAS are essential for inhibition of HR-DNA repair.**

A, B   Schematics of HR and NHEJ reporter assays.

C, D   Obtained results showing enhanced HR efficiency upon knockdown of endogenous cGAS in U2OS cells. Immunoblot inserts depict knockdown efficiency of cGAS and histone H1.2.

E, F   Results showing the effect of hcGAS, hcGASΔcGAMP, hcGASΔDNA, or hcGASΔOligo on HR (E) or NHEJ (F) in HEK293 cells. Corresponding immunoblot inserts depict cGAS expression.

Data information: Data are means with SEM, n = 3. Statistical significance was assessed using one-way ANOVA followed by Sidak's post-test. NS: $P > 0.05$, *$P ≤ 0.05$, ****$P ≤ 0.0001$.

Source data are available online for this figure.

localization by testing the impact of hcGAS-NLS and hcGAS-NES on HR-DNA repair. Consistent with its increased accumulation in the nucleus (Fig EV2C and D), cGAS-NLS had a stronger inhibitory effect on HR-DNA repair than hcGAS. In contrast, hcGAS-NES exhibited a weaker inhibitory effect (Fig EV4A)—consistent with its reduced presence in the nucleus (Fig EV2C and D). Thus, the ability of nuclear cGAS to impede HR-DNA repair depends on its DNA binding and oligomerization but not its

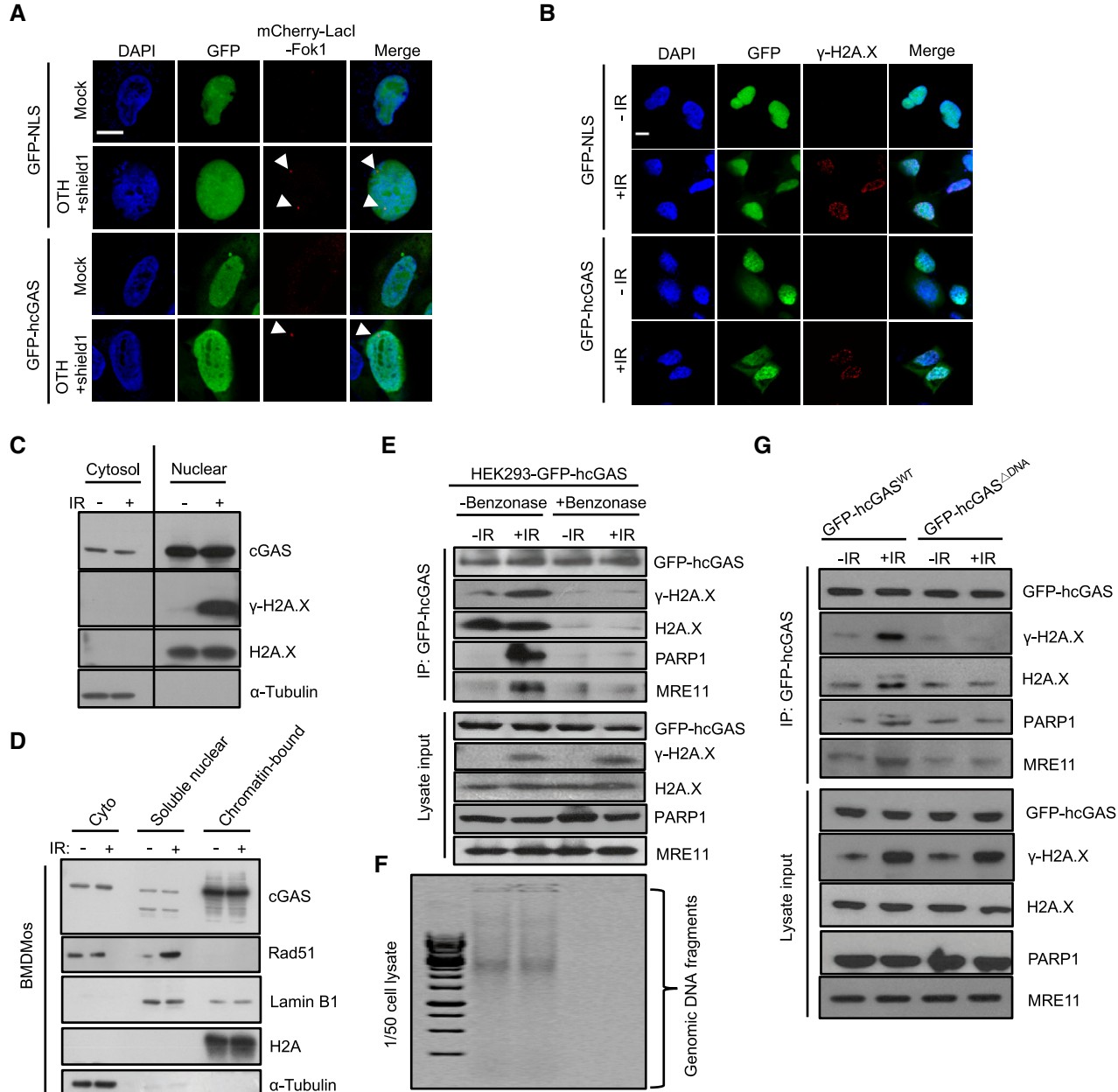

**Figure 6. Nuclear cGAS is constantly bound to the chromatin and not specifically recruited to DSB sites.**

A cGAS is not recruited to DSB sites: Confocal microscopic images of GFP-NLS- or GFP-hcGAS-expressing U2OS-DSB reporter cells incubated (or not) with Shield-1 and 4-OHT to induce the expression and translocation of mCherry-LacI-FokI (red) to specific DSB sites. Scale bar: 10 μm. The arrowheads indicate DSB sites.

B cGAS does not co-localize with γ-H2AX at DSB sites: GFP-NLS- or GFP-hcGAS-expressing HEK293 cells exposed (or not) to γ-irradiation (IR: 10 Gy), then stained for γ-H2AX. Scale bar: 10 μm.

C, D Nuclear cGAS is mainly chromatin-bound and remains unaltered upon γ-irradiation. (C) Cytosolic (cyto) and nuclear fractions of γ-irradiated (10 Gy, 30 min) BMDMos analyzed for cGAS and indicated molecules. (D) Cytosolic, soluble nuclear, and chromatin fractions from BMDMos were immunoblotted for cGAS and indicated proteins.

E–G cGAS co-isolates with DNA repair proteins because of bound chromatin bridges. (E) Nuclease digestion abrogates the co-isolation of cGAS and DNA repair proteins: Lysates of control (−IR) and γ-irradiated (+IR, 10 Gy, 30 min) GFP-hcGAS-expressing HEK293 cells were treated (or not) with benzonase before cGAS immunoprecipitation and analysis for indicated proteins. (F) Agarose gel analysis of DNA in corresponding cell lysates in (E). (G) Co-isolation of cGAS and DNA repair proteins depends on its binding to DNA: cGAS pulldowns along with lysate inputs of control and γ-irradiated HEK293 cells expressing GFP-hcGAS or GFP-hcGASΔDNA probed for indicated proteins.

Source data are available online for this figure.

enzymatic function. The positively charged cGAS N terminus of cGAS has recently been reported to promote DNA binding, formation of phase-separating complexes (Du & Chen, 2018), and association of cGAS with centromeric DNA (Gentili *et al*, 2019). However, the hcGAS^cat lacking the N-terminal domain was also found to inhibit HR-DNA repair (Fig EV4B), demonstrating that the N-terminal domain is dispensable for cGAS-mediated inhibition of HR-DNA repair.

### Nuclear cGAS is constantly bound to chromatin and not specifically recruited to DSB sites

To understand the specific signaling step in HR-DNA pathway targeted by cGAS, we asked whether cGAS affects activation of the upstream HR kinase ATM. However, expression of GFP-hcGAS or hcGAS^ΔcGAMP in HEK293T or HEK293 cells had no effect on γ-irradiation-induced phosphorylation of ATM (Fig EV4C and D). Similarly, WT, cGAS^−/−, and *Sting*^−/− BMDMos showed comparable γ-irradiation-induced ATM activation (Fig EV4E).

Recently, Liu *et al* (2018) similarly reported a role for cGAS in the regulation of HR. They proposed that in response to DNA damage, cGAS is actively imported from the cytosol to the nucleus and impedes HR via protein–protein interactions with PARP1 and H2A.X at DNA damage sites. To interrogate this model with our findings showing that cGAS is constantly present in the nucleus, we also examined whether cGAS is recruited to DSB sites. For this, we employed a DSB reporter system based on a mCherry-LacI-FokI nuclease fusion protein for DSB induction within a single genomic locus in U2OS cells (U2OS-DSB reporter) (Shanbhag *et al*, 2010). Although abundant in the nucleus, GFP-hcGAS did not show specific co-localization with mCherry-LacI-FokI at the DSB sites (Fig 6A). Similarly, GFP-hcGAS did not show specific co-localization with γ-H2A.X foci in γ-irradiated HEK293 cells (Fig 6B). Further, subcellular fractionation studies revealed that in the nucleus, endogenous mcGAS is mainly associated with chromatin prior to DNA damage (Fig 6C and D). To investigate further, we asked whether DNA damage was associated with specific interaction of cGAS with DNA repair proteins. Consistent with observations by Liu *et al*, GFP-hcGAS co-immunoprecipitated with γ-H2A.X, PARP1, and MRE11 in γ-irradiated HEK293 cells. However, such co-isolation dissipated when lysates were pre-treated with the nuclease benzonase (Fig 6E and F). This demonstrates that cGAS association with these DNA repair proteins occurs via nucleic acid as a bridge. Accordingly, in contrast to GFP-hcGAS, the DNA binding mutant GFP-hcGAS^ΔDNA did not show increased co-immunoprecipitation with γ-H2A.X, PARP1, or MRE11 upon γ-irradiation (Fig 6G). Thus, even though cGAS has been suggested to translocate from the cytosol to the nucleus and to interact with PARP1 and γ-H2A.X at DSB sites (Liu *et al*, 2018), our analysis provides no evidence to support these premises. Instead, we find that cGAS is constitutively present in the nucleus as a chromatin-bound protein and hence appears to interact with DNA repair proteins due to the recruitment of these factors to chromatin upon DNA damage. Together, the above data indicate that inhibition of DNA repair by chromatin-bound cGAS does not stem from defects in proximal signaling events at DSB sites.

### cGAS impedes HR by interfering with RAD51-mediated DNA strand invasion

The RAD51 recombinase acts downstream of ATM to catalyze HR-mediated DSB repair. Specifically, protomers of RAD51 form a protein filament on 3′ single-stranded DNA (ssDNA) tails stemming from the DSB end resection process. The RAD51-ssDNA filament, also referred to as the presynaptic filament, searches for and invades a homologous duplex target and exchanges ssDNA strands with the latter to generate a displacement loop (D-loop). This is followed by DNA synthesis and resolution of DNA intermediates to complete repair (Chapman *et al*, 2012; Ceccaldi *et al*, 2016). Given the above data indicating that cGAS-mediated HR inhibition was downstream of ATM, we then examined the effect of cGAS on RAD51 foci formation and downstream processes. GFP-hcGAS was found not to co-localize or affect RAD51 foci formation in γ-irradiated HEK293 cells (Fig 7A). Next, we inquired whether cGAS affects the downstream step involving dsDNA invasion by RAD51 filaments. To do this, we tested purified mcGAS^cat and hcGAS^cat on the RAD51-mediated D-loop reaction (Fig EV5A and B). Pre-incubation of the supercoiled dsDNA template with cGAS (schematics, Fig 7B (i), D) led to a strong inhibition of D-loop formation (Fig 7Ci and E compare lanes 2 versus 3–6). However, if RAD51 filaments were pre-bound to template dsDNA (schematic Fig 7B (ii)), then cGAS did not interfere with RAD51-mediated D-loop formation (Fig 7Cii, compare lane 7 versus 8). This demonstrates that cGAS-mediated attenuation of D-loop formation was not due to inhibition of the enzymatic activity of RAD51 but due to hindering invasion of dsDNA template by RAD51 filaments. Accordingly, when pre-incubated with linear dsDNA template, cGAS also inhibited DNA strand exchange mediated by human and yeast Rad51 protein (Fig EV5B–H). In both assay systems, inhibition occurred regardless of whether ATP and GTP, the precursors required for cGAS-mediated cGAMP synthesis, were present in the reaction, thus confirming that the inhibitory activity of cGAS is not cGAMP-mediated.

Next, we examined whether the observed inhibition of D-loop formation is a specific feature of cGAS as opposed to a universal feature of dsDNA binding proteins. To address this, we tested MHF, a component of the Fanconi anemia (FA) core complex (Zhao *et al*, 2014) that binds dsDNA with a similar affinity as cGAS alongside the latter (Appendix Fig S2A and B). In sharp contrast to cGAS, we found that MHF does not inhibit D-loop formation by human RAD51 (Appendix Fig S2C and D), demonstrating that the observed inhibition of D-loop formation is due to inherent features of cGAS and likely requires more than simple binding to dsDNA.

### cGAS impedes HR repair by compacting template DNA into a higher-ordered state resistant to RAD51-mediated DNA strand invasion

Compaction of DNA into higher-ordered state is a barrier to HR (Downs *et al*, 2003; Murga *et al*, 2007). By complexing with dsDNA, cGAS can form higher-ordered ladder-like or phase-separating structures (Andreeva *et al*, 2017; Du & Chen, 2018; Hooy & Sohn, 2018). In view of this, we asked whether cGAS-mediated inhibition of HR was via its compacting bound template dsDNA into a higher-ordered state less amenable to invasion by RAD51 filaments. To test this idea, we analyzed the ability of different cGAS mutants to form

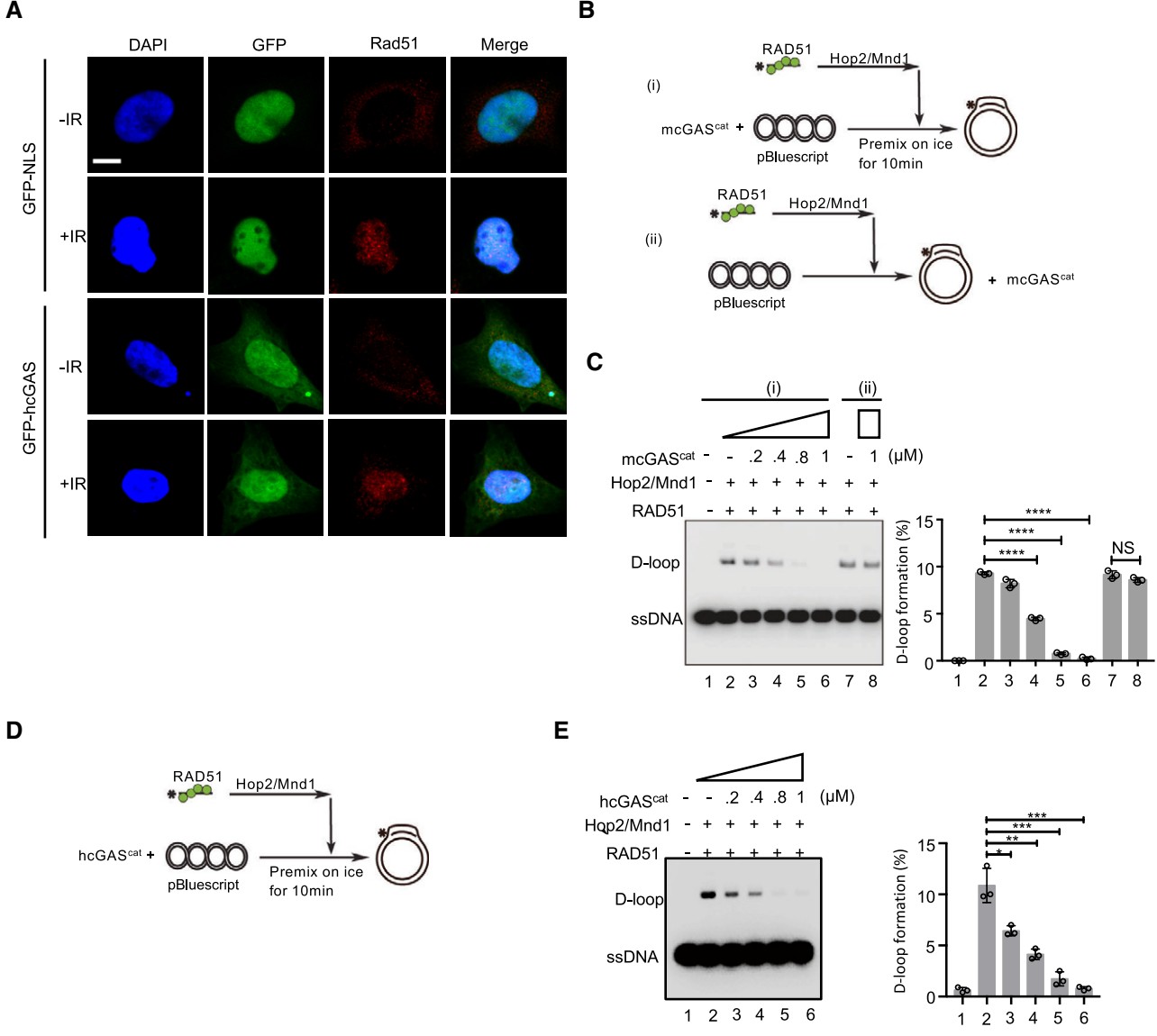

Figure 7. cGAS inhibits the HR-DNA repair by impeding RAD51-mediated strand invasion.

A Confocal images of γ-irradiated GFP-NLS- and GFP-hcGAS-expressing HEK293 cells stained for RAD51 (red) with or without γ-irradiation. Scale bar: 10 μm.
B Schematics of the D-loop formation assay, including pre-incubation of template dsDNA with cGAS^cat (i) or with cGAS^cat being added after RAD51 was bound to dsDNA (ii).
C Pre-incubation of dsDNA with mcGAS^cat prevents D-loop formation by human RAD51, but does not affect the RAD1 activity once RAD51 filaments are bound to dsDNA. The percentage of D-loop formed in each reaction (left) was graphed as the average of triplicates ± SD.
D Schematics of the D-loop assay.
E Pre-incubation of template dsDNA with hcGAS^cat blocks subsequent D-loop formation. The percentage of D-loop formation (below) was graphed as the average of triplicates ± SD.

Data information: Unpaired two-tailed Student's t-test was used for statistical analyses. NS P > 0.05, *P ≤ 0.05, **P ≤ 0.01, ***P ≤ 0.001, ****P ≤ 0.0001.
Source data are available online for this figure.

complexes and inhibit D-loop formation. To image dsDNA-cGAS complexes, we employed negative-stain electron microscopy (nsEM) (Hooy & Sohn, 2018). In agreement with our hypothesis and consistent with the above data, full-length (FL) hcGAS, hcGAS^cat-WT, and the catalytic dead hcGAS^cat-ΔcGAMP induced the formation of large cGAS-dsDNA complexes (Fig 8A) and inhibited RAD51-mediated D-loop formation (Fig 8B, C and I). This was in contrast to the

oligomerization hcGAS^cat-ΔOligo mutants that did not cluster dsDNA (Fig 8A) and was defective in inhibiting D-loop formation (Fig 8B, C and I).

In the recent study by Liu et al (2018), the authors reported that cGAS nuclear localization and subsequent HR inhibition were due to its active translocation from the cytosol and proposed that this was independent of its DNA binding. Further, they proposed

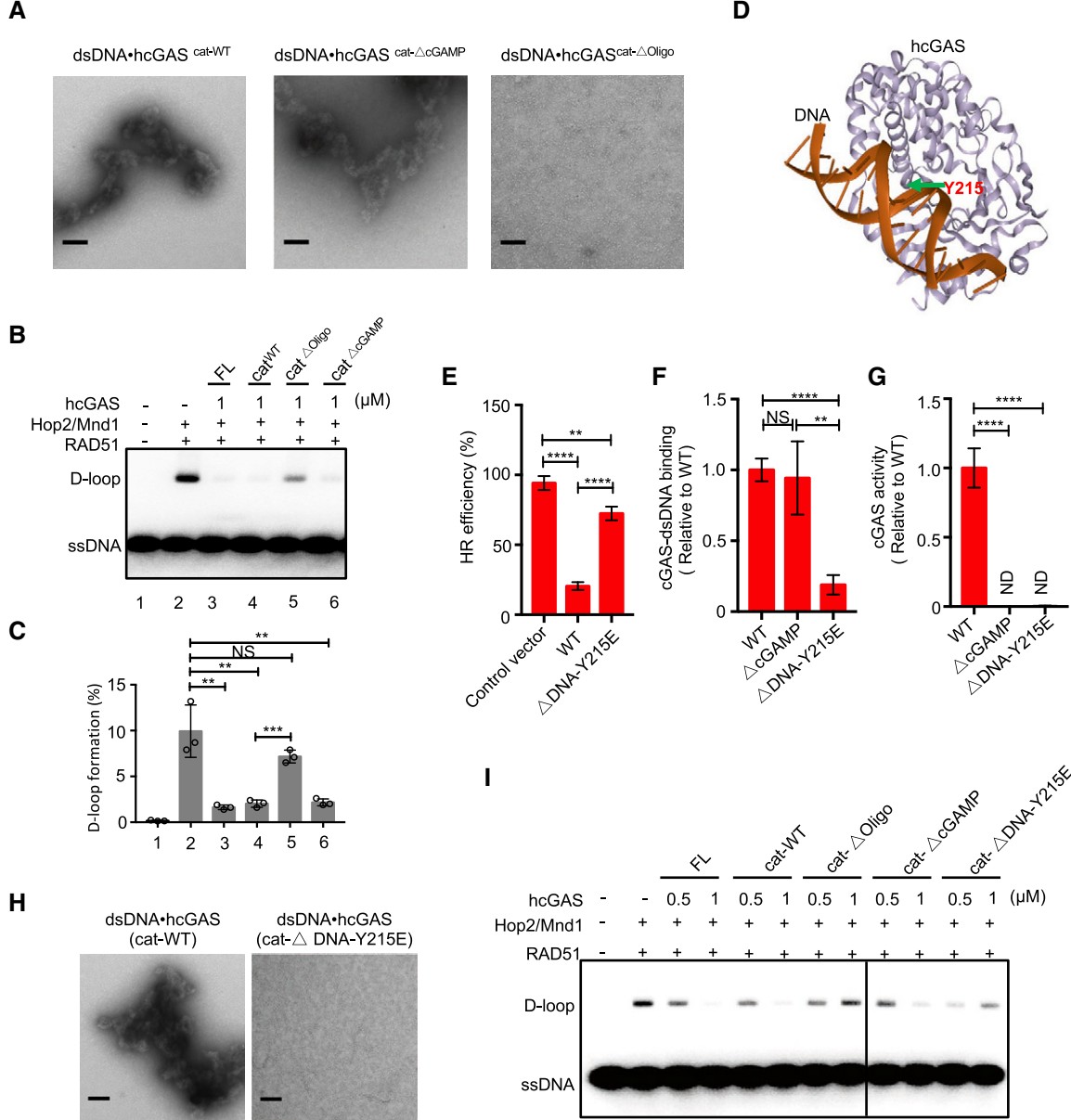

**Figure 8. cGAS compacts dsDNA and inhibits D-loop formation via oligomerization.**

A  Negative-stain electron micrographs of cGAS-dsDNA complexes following incubation of dsDNA with indicated cGAS variants. Scale bar: 100 nm.
B  Effect of indicated hcGAS variants on D-loop formation when pre-incubated with dsDNA.
C  Percentage of D-loop formed in each reaction (left) graphed as the average of triplicates ± SD.
D  Overview of a single 1:1 hcGAS-DNA complex depicting the location of the Y215 within the cGAS-dsDNA interface.
E  DR-GFP assay showing that hcGASΔDNA-Y215E is impaired in HR inhibition.
F  hcGASΔDNA-Y215E but not hcGASΔcGAMP has a decreased affinity to dsDNA24.
G  hcGASΔDNA-Y215E and hcGASΔcGAMP are defective in synthase activity.
H  Negative-stain electron micrographs showing that hcGAScat-ΔDNA-Y215E is defective in inducing cGAS-dsDNA complexes. Scale bar: 100 nm.
I  Effect of indicated hcGAS variants on D-loop formation.

Data information: Data are means ± SD, $n = 3$. Unpaired Student's *t*-test was used for statistical analyses: NS $P > 0.05$, **$P \leq 0.01$, ***$P \leq 0.001$, and ****$P \leq 0.0001$.
Source data are available online for this figure.

phosphorylation of cGAS at the conserved tyrosine 215 (Y215) as the mechanism by which cytosolic cGAS is prevented from translocating into the nucleus at steady state. The Y215 phosphorylation site is positioned within the cGAS-DNA interface (Fig 8D). To

independently verify the above observation and reconcile our observations with those by Liu *et al*, we mimicked tyrosine phosphorylation of cGAS by mutating Y215 to glutamic acid. Similarly, we found cGAS$^{\Delta DNA-Y215E}$ to be attenuated in HR repair inhibition (Fig 8E). To

elucidate further, we characterized purified cGAS$^{cat-\Delta DNA-Y215E}$. We found it to be impaired not only in DNA binding (Fig 8F) and synthase activity (Fig 8G) but also in its ability to compact dsDNA (Fig 8H) and inhibit D-loop formation (Fig 8I). Together with the above demonstration that cGAS nuclear localization and ability to compact dsDNA and inhibit D-loop formation require DNA binding, these results underscore the importance of cGAS-DNA interactions in HR inhibition and offer alternative interpretation for the recent observations (Liu *et al*, 2018).

## Discussion

Here, we have shown that cGAS is a chromatin-bound protein that restrains HR and that this function is independent of its enzymatic activity or the canonical STING-IFN-I pathway. Mechanistically, we show that cGAS hinders RAD51-mediated DNA strand invasion, a critical step in HR, and that this feature is linked to its ability to self-oligomerize, thereby compacting bound dsDNA into higher-ordered complexes resistant to invasion by the RAD51 filaments. This mode of HR regulation is perhaps analogous to that by proteins such as the linker histone H1 that also promotes DNA compaction and impedes invasion of homologous dsDNA template by RAD51 filaments (Downs *et al*, 2003; Hashimoto *et al*, 2007; Murga *et al*, 2007; Machida *et al*, 2014). This mechanism is different from that by Liu *et al* (2018), involving active translocation of cGAS from the cytosol into the nucleus to impede HR via specific interactions with DNA repair proteins including PARP1 and H2AX. In contrast, we find that cGAS is a chromatin-bound protein and that the co-isolation of cGAS with these proteins stems from indirect associations via bound chromatin bridges. Further, our analysis of cGAS mutants including the cGAS$^{\Delta DNA-Y215E}$ that mimics the phosphorylation proposed by Liu *et al* (2018) to control cGAS nuclear importation demonstrates that cGAS-DNA interactions are the principle for cGAS nuclear localization and HR-DNA repair inhibition.

What is the biological relevance of cGAS-mediated attenuation of HR? We posit that under homeostatic conditions, cGAS may generally function as a negative regulator to suppress undesirable genome rearrangements including chromosomal translocation, deletion, inversion, or loss of heterozygosity. On the other hand, by inhibiting HR in proliferating cells, as we have shown, cGAS accelerates genome destabilization and death of cells under acute genomic stress. While restricting the propagation of cells with defective genomes and therefore potentially cancerous, this cGAS function would also contribute to deleterious effects of DNA damage. For example, here we have shown that cGAS accelerates γ-irradiation-induced bone marrow ablation *in vivo*.

The demonstration herein that cGAS is chromatin-bound and is adept at compacting DNA into a higher-ordered state opens the door for further studies to examine whether, aside from its HR regulatory role, nuclear cGAS also affects other processes sensitive to changes in chromatin dynamics. An equally important issue is how the synthase activity of chromatin-bound cGAS is blocked to avert immunopathology. Multiple concurrent mechanisms are likely in play. One example is the recently reported circular RNA cia-cGAS that binds and blocks cGAS (Xia *et al*, 2018). Additionally, we posit that in contrast to naked dsDNA, binding to dsDNA in the context of the complex chromatin matrix may be unfavorable for cGAS

synthase activity. It is also possible that the inactivity of nuclear cGAS might in part be due to post-translational modifications yet to be elucidated.

Dysregulations in the DNA damage response and the immune system are at the core of many human afflictions including infections, autoimmunity, neurodegeneration, cancer, and aging-associated disorders. cGAS has been implicated in many of these health conditions. The dual function of cGAS in the cytosol as an innate immune sensor and a negative regulator of DNA repair in the nucleus underscores the central position of cGAS in the cross-talks between the innate immune system and the DNA damage response. This work opens new avenues for future research into how the cytosolic and nuclear functions of cGAS are regulated and their impact on health and disease. For instance, because cGAS promotes both tumorigenesis (Ahn *et al*, 2014; Dou *et al*, 2017; Bakhoum *et al*, 2018) and anti-tumor immunity (Deng *et al*, 2014; Woo *et al*, 2014; Harding *et al*, 2017; Wang *et al*, 2017), pinpointing the extent and biological context in which the distinct subcellular functions of cGAS contribute to these processes, and how these cGAS functions may be manipulated, will be beneficial for achieving the desired outcome of DNA damage- and immune-based anti-tumor therapies.

## Materials and Methods

### Mice

All mice in this study were on C57BL/6 background. *Sting$^{-/-}$* (C57BL/6J-Tmem173gt/J) (Sauer *et al*, 2011) and *cGAS$^{-/-}$* (B6(C)-Mb21d1tm1d (EUCOMM) Hmgu/J) (Schoggins *et al*, 2014) were from Jackson Laboratory. *cGAS$^{-/-}$ Sting$^{-/-}$* mice were generated by interbreeding *cGAS$^{-/-}$* with *Sting$^{-/-}$* mice. Mice were bred in specific pathogen-free animal facility of Umeå Center for comparative Biology (UCCB), and experiments were carried out according to the guidelines set out by the Umeå Regional Animal Ethic Committee (Umeå Regionala Djurförsöksetiska Nämnd, Approval No. A53-14).

### Bone marrow depletion

For irradiation, WT, *cGAS$^{-/-}$*, *Sting$^{-/-}$*, and *cGAS$^{-/-}$ Sting$^{-/-}$* mice were anesthetized by intraperitoneal injection of 150 µl of a mixture containing 8% Ketaminol vet. (Intervet AB, Sollentuna, Sweden) and 5% Dormitor vet. (Orion Pharma AB, Espoo, Finland). Mice were the placed in a Gammacell 40 Irradiator (MDS Nordion) with a 137Cs gamma-ray source. Radiation was given as a single dose of 1 Gy per min for 9 min (total dose of 9 Gy). At specified time points after irradiation, mice were sacrificed and bone marrow cells were isolated, counted, and analyzed by flow cytometry for the following populations hematopoietic progenitor cells (cKit$^+$Sca-1$^+$), B cells (B220$^+$), and neutrophils (Gr1$^+$Ly6G$^+$). The total bone marrow cells or specified cell populations in the femur were calculated and expressed as relative (percentage) to non-irradiated WT controls. Our FACS gating strategy is depicted in Appendix Fig S3.

### Antibodies and reagents

The anti-α-tubulin antibody, aphidicolin, and nocodazole were purchased from Sigma-Aldrich. The anti-p-ATM (Ser1981), cGAS,

and GFP antibody were from Santa Cruz. Antibodies against ATM, Flag, mouse cGAS, human cGAS, STING, MRE11, PARP1, H2A, H2A.X, γ-H2A.X p-IRF3, and IRF3 were from Cell Signaling Technology; Alexa 488-anti-Sca-1 was from Invitrogen and PECY7-anti-cKit; V450-Ly6G and FITC-anti-GR1 were from BD Pharmingen; 2′,3′-cGAMP and immunostimulatory DNA (ISD) were from InvivoGen; and ATP was from New England Biology, while GTP, Rad51, and Lamin B1 antibody were from Abcam.

## Plasmid and construct cloning

pTRIP-SFFV-EGFP-NLS(GFP-NLS), pTRIP-CMV-GFP-FLAG-hcGAS (GFP-hcGAS), and pTRIP-CMV-GFP-FLAG-hcGAS E225A-D227A (GFP-hcGAS(ΔcGAMP) (Addgene plasmid #86677, #86675, and #86674) have been described previously (Raab *et al*, 2016). pTRIP-CMV-GFP-FLAG-hcGAS C396A-C397A(GFP-hcGAS(ΔDNA)) and pTRIP-CMV-GFP-FLAG-hcGAS K394E were generated by site-directed mutagenesis from pTRIP-CMV-GFP-FLAG-hcGAS(GFP-hcGAS). Flag-hcGAS was cloned into pcDNA3.1+ to generate the pcDNA-hcGAS plasmid. To generate pcDNA-hcGAS$^{cat}$, the C-terminal aa161-522 of hcGAS was PCR-amplified and cloned into pcDNA. pcDNA-hcGAS E225A-D227A(pcDNA-hcGAS(ΔcGAMP), pcDNA-hcGAS C396A-C397A(GFP-hcGAS(ΔDNA), and pcDNA-hcGAS K394E were generated by site-directed mutagenesis from the pcDNA-hcGAS plasmid. The SV40 NLS (nuclear localization signal) sequence (5′-CCAAAAAAGAAGAGAAAGGTA-3′) was cloned separately into C-terminal of pTRIP-CMV-GFP-FLAG-hcGAS and pcDNA-hcGAS to generate pTRIP-CMV-GFP-FLAG-hcGAS-NLS and pcDNA-hcGAS-NLS. NES (nuclear export signal) sequence (5′-CTG CCCCCCCTGGAGCGCCTGACCCTG-3′) was cloned separately into C-terminal of pTRIP-CMV-GFP-FLAG-hcGAS and pcDNA-hcGAS to generate pTRIP-CMV-GFP-FLAG-hcGAS-NES and pcDNA-hcGAS-NES. pHPRT-DR-GFP and pCBASceI were gifts from Maria Jasin (Addgene plasmid # 26476 and # 26477) (Pierce *et al*, 2001). pimEJ5GFP was a gift from Jeremy Stark (Addgene plasmid # 44026) (Bennardo *et al*, 2008).

## Cells and cell culture

HEK293, HEK293T, U2OS, HeLa, Raw 264.7, and THP-1(ATCC) cells were cultured under 5% $CO_2$ at 37°C in Dulbecco's modified Eagle's medium (DMEM, high glucose, GlutaMAX) (Life Technologies) containing 10% (v/v) fetal calf serum (FCS, Gibco) and 1% (v/v) penicillin (100 IU/ml)+streptomycin (100 μg/ml). Bone marrow-differentiating monocytes (BMDMos) were generated by culturing the mouse bone marrow cells in IMDM (Gibco, Life Technologies) supplemented with 10% (v/v) FCS (Gibco, Life Technologies), 1% (v/v) penicillin (100 IU/ml)/streptomycin (100 μg/ml), 2 mM glutamine (Sigma-Aldrich), and 20% (v/v) L929 conditional medium and maintained with 5% $CO_2$ at 37 °C. The cells were used for experiment on 4 days after start of differentiation. Bone marrow-derived macrophages (BMDMs) were used 7–10 days after the start of differentiation.

## Generation of stable overexpression cell lines

HEK293T cells were transfected with psPAX2, pMD2.G plasmids, and the lentiviral vector pTRIP containing the open reading frame of GFP-NLS or GFP-cGAS or GFP-cGAS mutants by using Lipofectamine LTX. The supernatants containing lentiviral particles were harvested at 48 h. HEK293 and HEK293T cells were then transduced with the lentiviral vectors by directly adding supernatant together with polybrene (5 μg/ml) to cells. 2 days later, GFP-positive cells were sorted by flow cytometry and propagated further. To generate stable HEK293T DNA damage reporters, HEK293T cells were transfected separately with pHPRT-DR-GFP (to monitor HR) and pimEJ5GFP (to monitor NHEJ), and 2 days later, cells were put under puromycin (2 μg/ml) selection. Single clones were picked and expanded for the reporter assays.

## Immunofluorescence

Cells were seeded and cultured on glass coverslips in 12-well plate and fixed in 4% paraformaldehyde (PFA) in PBS for 20 min at room temperature. Cells were permeabilized in 0.5% Triton X-100 for 10 min. Slides were blocked in 5% normal goat serum (NGS) and incubated with primary antibodies diluted in 1% NGS overnight at 4°C. Samples were then incubated with secondary antibodies labeled with Alexa Fluor 488 (Invitrogen) diluted in 1% NGS at RT for 1 h.

Thereafter, they were stained with DAPI (or plus Phalloidin) for 15 min at room temperature. Coverslips were mounted using Dako Fluorescence Mounting Medium (Agilent) and imaged using Nikon confocal (Eclipse C1 Plus). All scoring was performed under blinded conditions.

For quantification of the nuclear cGAS percentage, we used ImageJ software to quantify the nuclear cGAS immunofluorescence intensity relative to whole-cell cGAS intensity from 6 different fields with $n > 50$ cells. Percentage of nuclear cGAS = (nuclear cGAS/whole-cell cGAS intensity) X100%.

## Subcellular fractionation and immunoblotting

Cytoplasmic and nuclear extracts were prepared using the nuclear extraction kit (Abcam) according to the manufacturer's instructions. For chromatin-bound fraction, we use the Subcellular Protein Fractionation Kit (Thermo Fisher) according to the manufacturer's instructions. For other assays, cells grown in culture were trypsinized, pelleted, washed, and resuspended in a mild Nonidet P-40 lysis buffer (1% NP-40, 50 mM Tris–HCl, 150 mM NaCl, pH 7.5, 1 mM NaF, 2 mM PMSF, protease inhibitor cocktail [Roche Applied Science], 1 mM sodium orthovanadate, and 10 mM sodium pyrophosphate). The lysates were centrifuged at 10,000 *g* for 15 min, and proteins in supernatants were quantified by BCA reagent (Thermo Fisher Scientific, Rockford, IL). Proteins were resolved in SDS–PAGE, transferred to nitrocellulose membrane (Amersham Protran 0.45 μm NC), and immunoblotted with specific primary antibodies followed by HRP-conjugated secondary antibodies. Protein bands were detected by SuperSignal West Pico or Femto Chemiluminescence Kit (Thermo Fisher Scientific).

## Cell cycle analysis

Following the individual treatments (i.e., nocodazole treatment, aphidicolin treatment, and serum starvation), cells were washed twice in PBS, then fixed in cold 70% ethanol for 30 min at 4°C.

Thereafter, they were washed and treated with RNase to remove RNA. After washing, cells were stained with DAPI at 4°C. Flow cytometry was performed on BD LSR II flow cytometer, and the data were analyzed with FlowJo software.

### DNA damage-induced cell death

BMDMos were synchronized at G2/M by incubation with 100 nM nocodazole for 12 h. Thereafter, they were γ-irradiated, then released and evaluated for cell death at indicated time points. Irradiation-induced cell death was determined by XTT assay (Sigma-Aldrich) according to the manufacturer's instructions. Absorbency was measured with a spectrophotometer (Tecan Infinite M200 Microplate Reader) at 450 nm with a reference wavelength at 650 nm. Relative number of dead cells as compared to the number of cells without treatment was expressed as percent cell death using the following formula: cell death (%) = 100% − 100% X(A450 of treated cells/A450 of untreated cells).

### Comet assay

Cells were γ-irradiated in a 137Cs gamma-ray source (Gammacell 40 irradiator, MDS Nordion) with indicated dose, and chromosome fragmentation was determined by comet assay. Briefly, during irradiation all the cells are kept in ice to stop the DNA repair process. Thereafter, cells were transferred to 37°C to allow DNA repair and then harvested at indicated time points for analysis. $1 \times 10^5$ cells/ml in cold PBS were resuspended in 1% low-melting agarose at 40°C at a ratio of 1:3 vol/vol and pipetted onto a CometSlide. Slides were then immersed in prechilled lysis buffer (1.2 M NaCl, 100 mM EDTA, 0.1% sodium lauroyl sarcosinate, 0.26 M NaOH, pH > 13) for overnight (18–20 h) lysis at 4°C in the dark. Slides were then carefully removed and submerged in room temperature rinse buffer (0.03 M NaOH and 2 mM EDTA, pH > 12) for 20 min in the dark. This washing step was done 2 times.

Slides were transferred to a horizontal electrophoresis chamber containing rinse buffer and separated for 25 min at voltage 0.6 V/cm. Finally, slides were washed with distilled water and stained with 10 μg/ml propidium iodide and analyzed by fluorescence microscopy. Twenty fields with about 200 cells in each sample were evaluated and quantified by the Fiji software to determine the tail length (tail moment).

### Pulsed-field gel electrophoresis

BMDMos from WT and $cGAS^{-/-}$ mice were irradiated (20 or 30 Gy) on ice (time 0), then incubated at 37°C to allow DNA repair. At different time points post-irradiation (15, 45 min), cells were washed twice with ice-cold phosphate-buffered saline (PBS). Cell pellets (plugs) were immediately placed in 10× volume of lysis buffer (0.5 M EDTA (pH 9.5), 1% sarkosyl, and 1 mg/ml proteinase K (Sigma) for a 48-h digestion at 50°C with one buffer change after 24 h. Following lysis, the plugs were washed for at least 24 h with 10× volume of TE buffer containing 10 mM phenylmethylsulfonyl fluoride (PMSF) at room temperature. After PMSF treatment, the plugs were washed three times for at least 2 h in each case in 10× volume of TE buffer without PMSF at room temperature.

Electrophoresis was performed in a CHEF-DR II apparatus (Bio-Rad) with a hexagonal array of 24 electrodes, which produce a field reorientation angle of 120°. The plugs were inserted into 1% gels made from high tensile strength agarose (pulsed-field grade agarose; Bio-Rad) in 0.5× TBE. The gel was run at 13°C in 0.5× TBE (the buffer was recirculated through a refrigeration unit to keep the temperature constant and to avoid ion buildup at the electrodes) for 36 h. The pulse time was increased during the run linearly from 50 to 150 s at a field strength of 6 V/cm. After electrophoresis, the gels were stained for 1 h in 200 ml staining buffer (TE containing 10 μg/ml ethidium bromide) and de-stained for 3 h in the same buffer in the absence of ethidium bromide. After that, the signals were detected by a Gel-DOC System.

### Determination of micronuclei

HEK293 cells were exposed (or not) to γ-irradiation and cultured for 24 h. BMDMos arrested at G2 by incubating with nocodazole were exposed to γ-irradiation, then released and cultured for 48 h. Cells were fixed, permeabilized in 0.5% Triton X-100, stained with the DNA dye DAPI, then analyzed by microscopy for the presence of micronuclei. Micronuclei were defined as discrete DNA aggregates separated from the primary nucleus in cells where interphase primary nuclear morphology was normal. Cells with an apoptotic or necrotic appearance were excluded.

### HR and NHEJ reporter assays

Homologous recombination (HR) and NHEJ repair in HEK293T cells were measured as described previously using the DR-GFP stable cells (Pierce et al, 2001) and EJ5-GFP stable cells (Bennardo et al, 2008). Briefly, $0.5 \times 10^6$ HEK293T stable reporter cells were seeded in 6-well plates co-transfected with 2 μg I-SceI expression plasmid (pCBASce) and either 4 μg pcDNA-hcGAS mutants or empty pcDNA vector. Forty-eight hours post-transfection, cells were harvested and analyzed by flow cytometry analysis for GFP expression. Means were obtained from three independent experiments. U2OS cells silenced for using cGAS were transfected with 2 μg I-SceI expression plasmid (pCBASce) for 2 days, then harvested and analyzed by flow cytometry analysis for GFP expression.

### Protein purification

Recombinant hcGAS constructs (full-length or catalytic domains (cGAS$^{cat}$)) were purified as described in Hooy and Sohn (2018). Briefly, hcGAS constructs were cloned into the pET28b vector (Novagen) with an N-terminal MBP-tag and a TEV protease cleavage site. Protein expression was induced using 200 μM IPTG at 16°C for overnight in E. coli BL21 Rosetta 2, then purified by amylose affinity chromatography, cation-exchange chromatography, and size-exclusion chromatography. Tag-free, purified cGAS proteins were then stored in 80°C with a buffer containing 20 mM Tris–HCl at pH 7.5, 300 mM NaCl, 10% glycerol, 5 mM DTT. Purified mouse cGAS (mcGAS aa 141–507) was a gift from Karl-Peter Hopfner, and the method for purification has been described (Andreeva et al, 2017). Human RAD51, Hop2/Mnd1, MHF, and budding yeast Rad51 and Rad54 were purified as described previously (Sung, 1994; Petukhova et al, 1998; Chi et al, 2007; Zhao et al, 2014).

### cGAS enzymatic activity and DNA binding assays

cGAS activity was assayed using the pyrophosphatase-coupled assay (Seamon & Stivers, 2015) with modifications. Briefly, cGAS was incubated with 50 nM E. coli pyrophosphatase, equimolar concentrations of ATP, and GTP plus dsDNAs in the reaction buffer. At different time points, aliquots were collected and mixed with an equal volume of quench solution (reaction buffer minus $Mg^{2+}$ plus 25 mM EDTA). Quenched solutions were then mixed with 10 μl malachite green solution and incubated for 45 min at RT. Absorbance at ~620 nm was compared to an internal standard curve of inorganic phosphate to determine the concentration of phosphate in each well. Phosphate concentrations of control reactions devoid of recombinant cGAS were subtracted from reactions containing recombinant cGAS. Apparent catalytic rates were calculated from the slopes of control-subtracted phosphate concentrations over time. Reported rates were halved to reflect pyrophosphate production. The activity of catalytic cGAS domain mutants was expressed as relative to wild-type cGAS$^{cat}$ (cGAS$^{cat-WT}$).

DNA binding to plasmid DNA was analyzed by DNA mobility shift assay. Briefly, cGAS (20–200 nM) or MHF hetero-tetramer (Zhao *et al*, 2014) (20–200 nM) was incubated with an 80-mer double-strand DNA substrate (dsDNA, 10 nM each) at 37°C for 10 min in 10 μl of buffer B (25 mM Tris–HCl, pH 7.5, 1 mM DTT, 100 μg/ml BSA, 1 mM MgCl₂, and 45 mM KCl). The reaction mixtures were resolved in 7% polyacrylamide gels in TAE buffer (40 mM Tris, 20 mM Acetate and 1 mM EDTA) at 4°C. Gels were dried onto Whatman DE81 paper (Whatman International Limited) and subject to phosphorimaging analysis. cGAS binding to dsDNA (24 bps) was assayed previously described (Hooy & Sohn, 2018). Briefly, increasing concentrations of cGAS constructs were added to a fixed concentration of fluorescein amidite-labeled (FAM) dsDNA (5–10 nM final). Changes in fluorescence anisotropy were plotted as a function of cGAS concentration and fit to the Hill equation. The binding affinity of catalytic cGAS domain mutants was expressed as relative to wild-type cGAS$^{cat}$ (cGAS$^{cat-WT}$).

### Negative-stain Electron Microscopy (nsEM)

Experiments were conducted as described previously (Hooy & Sohn, 2018). Briefly, cGAS:dsDNA complexes were prepared by incubating 200 nM recombinant cGAS$^{cat}$ wild-type or variants with 66 nM dsDNA564 for 15 min at 25 ± 2°C. Solutions were applied to glow-discharged, carbon-coated EM grids for 2 min, blotted, then stained with 1% uranyl acetate for 30 s, twice. Excess stain was aspirated, and the grid was allowed to dry at ambient temperature. Images were collected on a Phillips BioTwin CM120 (FEI) at the Johns Hopkins University School of Medicine.

### D-loop formation

The D-loop reaction was conducted as described previously (Raynard & Sung, 2009). Briefly, cGAS protein (0.2–1.0 μM) was pre-incubated with pBluescript dsDNA (36 μM base pairs) on ice for 10 min. Human RAD51 (0.6 μM) was incubated with $^{32}$P-labeled 90-mer ssDNA (2.4 μM nucleotides) at 37°C for 10 min to allow RAD51 filament formation. Hop2/Mnd1 complex (0.5 μM) was then added to the mixture, followed by a 2-min incubation at 37°C. The reaction was initiated by adding the cGAS-pBluescript dsDNA mixture and further incubated at 37°C for 5 min. The reaction mixtures were deproteinized before being resolved in 0.9% agarose gels in TBE buffer. Gels were dried, and the radiolabeled DNA species were revealed and quantified by phosphorimaging analysis. cGAS protein was also added after D-loop formation, followed by a further 5-min incubation at 37°C.

### DNA strand exchange assay

The assay was conducted at 37°C, and reaction mixtures were resolved by electrophoresis in non-denaturing 10% polyacrylamide gels in TAE buffer (45 mM Tris-acetate, pH 7.5, 0.5 mM EDTA) as described previously (Zhao & Sung, 2015). Briefly, the 150-mer oligo (6 μM nucleotides, P1 in Appendix Table S1) was incubated with human RAD51 (2 μM) in 10 μl of buffer G (25 mM Tris–HCl, pH 7.5, 60 mM KCl, 1 mM DTT, 100 μg/ml BSA, 1 mM ATP/1 mM GTP, and 2 mM MgCl₂) containing an ATP-regenerating system consisting of 20 mM creatine phosphate and 20 μg/ml creatine kinase for 5 min. cGAS was pre-mixed with $^{32}$P-labeled homologous dsDNA (6 μM base pairs, P2/P3 in Appendix Table S1) on ice for 10 min. The two reaction mixtures were combined, followed by the addition of 4 mM spermidine hydrochloride to 12.5 μl final volume. After 30 min of incubation, the reactions were stopped by adding an equal volume of 1% SDS containing proteinase K (1 mg/ml) and a 5-min incubation. Gels in which the deproteinized reaction mixtures had been resolved were dried and subject to phosphorimaging analysis.

### RT–qPCR

Total RNA was extracted using the TRIzol (Thermo Fisher) according to the manufacturer's protocol. cDNA was prepared using Maxima H Minus First-Strand cDNA Synthesis Kit and random oligomer primers (Thermo Fisher Scientific). qRT–PCR was performed using SYBR Select Master Mix (Thermo Fisher Scientific) on an QuantStudio 5 Real-Time PCR System (Thermo Fisher). The *IFNB1* transcript levels were normalized to the housekeeping gene 18S rRNA. The oligonucleotides used in thus study are depicted in Appendix Table S1.

### siRNA-mediated gene silencing

To silence cGAS, U2OS cells were transfected with a pool of the following siRNA from Thermo Fisher (sicGAS-1: 5′-GGAAG AAAUUAACGACAUU-3′; sicGAS-2: 5′-GAAGAAACAUGGCGGCUAU-3′; siH1.2-1: 5′-CGGCCACUGUAACCAAGAA-3′; siH1.2-2: 5′-GAAGAGC GCUAAGAAAACA-3′).

### FokI induced double-strand break system

The U2OS-FokI DSB reporter cells contain a stably integrated LacO array and a mCherry-LacI-FokI fusion protein fused to a destabilization domain (DD) and a modified estradiol receptor (ER) (ER-mCherry-LacI-FokI-DD) (Shanbhag *et al*, 2010). This enables inducible nuclear expression of ER-mCherry-LacR-FokI-DD after administration of the small-molecule Shield-1 ligand (stabilizes the

DD-domain) and 4-hydroxytamoxifen (4-OHT; induce nuclear translocation of ER-mCherry-LacR-FokI-DD). To induce site-specific double-strand breaks by FokI, these cells were incubated with 1 μM Shield-1 (cat. no. 632189, Clontech) and 1 μM 4-OHT (cat. no. H7904, Sigma-Aldrich) for about 5 h.

### Crystal structure of hcGAS.DNA complex

Crystal structure image of hcGAS-DNA complex was created with NGL Viewer from RCSB Protein Data Bank with accession numbers 6CTA (Rose *et al*, 2018; Zhou *et al*, 2018).

**Expanded View** for this article is available online.

### Acknowledgements

We are grateful to Karl-Peter Hopfner, Ludwig-Maximilians-University, Munich, for providing recombinant murine cGAS and Roger Greenberg, University of Pennsylvania, for the U2OS-DSB reporter cells. This work was funded by the Laboratory for Molecular Infection Medicine Sweden (MIMS), the Medical Faculty, Umeå University, the Swedish Research Council (grants 2015-02857 and 2016-00890 to N.O.G), the Swedish Cancer Foundation (grant, CAN 2017/ 421 to N.O.G), and the HHS|National Institutes of Health (NIH) (grant R01 CA220123 to P.S., grant P30 CA054174 to the Mays Cancer Center at the University of Texas Health Science Center at San Antonio, and grant R01 GM GM 129342-01-A1 to J.S.).

### Author contributions

HJ and NOG conceived the study. HJ, SP, XX, FL, JS, PS, and NOG, designed experiments and interpreted data. HJ, SP, XX, AK, FL, RMH, and NOG performed experiments. NOG supervised the research and together with HJ wrote the paper which other authors commented on.

### Conflict of interest

The authors declare that they have no conflict of interest.

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
