## [Review Process File · The EMBO Journal]

Chromatin-bound cGAS is an inhibitor of DNA repair hence accelerates genome destabilization and cell death

Hui Jiang, Xiaoyu Xue, Swarupa Panda, Ajinkya Kawale, Richard M. Hooy, Fengshan Liang, Jungsan Sohn, Patrick Sung and Nelson O. Gekara.

Review timeline:

Submission date:	18 th June 2019
Editorial Decision:	12 th July 2019
Revision received:	5 th August 2019
Editorial Decision:	12 th August 2019
Revision received:	19 th August 2019
Accepted:	2 nd September 2019

Editor: Hartmut Vodermaier

Transaction Report:

1st Editorial Decision

12th July 2019

Thank you again for submitting your nuclear cGAS modulating homologous recombination repair to The EMBO Journal. We have now received the below reports from two expert referees, who both consider your demonstration of continuous cGAS presence on chromatin and role in accelerating genome destabilization and cell death highly interesting and generally well-demonstrated. We shall therefore be happy to offer publication after addressing of a limited number of specific issues noted by the reviewers. As you will see, referee 1 raises three points regarding the DNA repair/recombination assays, of which especially point (a) should be experimentally addressed, and points (b) and (c) requiring at least further clarification and potential re-interpretation (although any additional data to decisively sort out these issues would certainly be helpful!). For referee 2, it will be important to directly address major points 3-5, while commenting/discussing should be sufficient for points 1 and 2.

REFeree REPORTS

Referee #1:

Chromatin-bound cGAS is an inhibitor of DNA repair hence accelerates genome destabilization and cell death

cGAS is an immune receptor that recognizes cytosolic DNA leading to a potent immune response via STING-IRF3-type I IFN signaling. Recognition of aberrant chromatin in micronuclei by cGAS links genome instability to the innate immune response. Genome integrity is ensured by accurate repair of DNA double-stranded breaks. Here Jiang et al. show that cGAS is constantly present in the nucleus in a chromatin-bound state, while acting as a negative regulator of HR-mediated DNA repair leading to increased micronuclei formation and cell death.

Overall this work is interesting and expands our understanding of the role of chromatin-associated nuclear cGAS. The data demonstrating that cGAS is constitutively present in the nucleus and cytosol is convincing and demonstrated by microscopy and clean fractionation assays. Especially exciting are the findings that next to cGAS immune activation via micronuclei, cGAS plays a crucial role in accelerating genome destabilization and cell death in vitro and in vivo. These important data open up the question how nuclear cGAS can promote micronuclei formation upon DNA damage. Jiang et al. suggest a mechanism, where constitutively chromatin-bound cGAS is directly impairing access of DNA repair factors, such as RAD51, to the damaged DNA. Overall, the data are solid, and the conclusions generally supported by the experimental data. I am very favorable for publishing this study and only have a few points:

Experimental points:

- a) The authors perform biochemical assays to reveal the mechanism behind HR-mediated DNA repair inhibition using the cGAS catalytic domain (Figure 6). In contrast, cell assays are based on full-length cGAS. To have these two sets of data comparable, it would be important to see if the cGAS catalytic domain is sufficient to inhibit DNA repair in cells as well.
- b) Specific blocking of D-loop formation by cGAS in vitro (Fig. S8). I find the conclusions of these experiments a bit premature. First, the conclusions of the authors that cGAS and MHF have similar binding affinities is not entirely justified, because the presented data may just well reflect a titration of binding sites (thus underestimating the affinity). In fact, the curve in Fig. S8 looks a bit like a linear slope immediately followed by a plateau which is characteristic for a titration experiment of a tight binder. If the authors want to make that conclusion, perhaps they could use a more quantitative, equilibrium assay such as fluorescence polarization anisotropy?
- c) Specific blocking of D-loop formation by cGAS in vitro (Fig. S8). Along the same lines. I find it rather surprising that MHF does not block D-loop formation at all at a concentration where it robustly binds DNA. Shouldn't there be at least a reduction simply due to competition with Rad51 for the dsDNA? Perhaps the conclusion is the other way around, i.e. MHF specifically allows D-Loop formation even though it is bound by DNA, while cGAS is a mere steric competitor? To draw the conclusion that cGAS "specifically" inhibits D-loop is not yet justified by these experiments.

Editorial:

- The introduction is quite brief and more background on the topics concerning this study would be highly beneficial to the general readership.
- Also, rationales behind experiments are sometimes lacking, for instance Figure 1a-c. What is the rationale of studying low and high density cultures and +/- serum?
- "DNA binding zinc finger" C396A/C397A mutant: please rephrase since the effect of the mutant is loss of DNA binding and it is a bit confusing as written.
- P6 "due to the recruitment of these factor to chromatin upon DNA damage". These "factors"?
- Fig. S8E typo: "Proetin"

Referee #2:

This manuscript sets out by reporting that a fraction of the cGAS protein - widely believed to be a cytosolic DNA sensor - is actually localised in the cell nucleus. Similar findings have been published recently by others, but these studies generally fell short of providing an explanation for nuclear functions of cGAS. Here, the authors show striking new data from experiments using ionising radiation (IR): cGAS-deficient cells, as well as cGAS knock-out mice, are less susceptible to IR. In cells lacking cGAS, IR induces less DNA damage and less cell death. In cGAS KO mice, IR fails to ablate the bone marrow. These effects are independent of STING and of the enzymatic activity of cGAS to produce cGAMP. The authors go on to show that cGAS blocks homologous recombination DNA repair, by blocking RAD51-coated DNA filaments from invading complementary DNA helices. Overall, the data are convincing, although this reviewer lacks detailed expertise in DNA repair. There are a number of interesting outstanding questions, most notably pertaining to the issue why cGAS is not activated to produce cGAMP under homeostatic conditions if constantly bound to nuclear DNA. However, these questions are probably best addressed in a

separate study. Indeed, the manuscript already reports a wealth of timely and important results that deserve swift publication in EMBO Journal, if the specific comments detailed below can be addressed.

Major points

1. Are cGAS-deficient cells also more resistant to other genotoxic stresses, apart from IR? Such data would broaden the study.
2. The authors suggest that cGAS is bound to chromatin "constantly" (Fig 5). Most of the evidence for this comes from experiments with overexpressed, GFP-tagged cGAS. Have the authors estimated how much endogenous cGAS protein molecules a cell contains, relative to the size of its DNA genome? Is cGAS essentially behaving like histones, binding the whole genome? It seems hard to imagine there is enough cGAS protein expressed. Alternatively, does cGAS bind specific regions in the genome, as suggested in PMID 30811988? At a minimum, this reference and PMIDs 30270045 & 28738408, which report assays on chromatin binding by cGAS, should be discussed.
3. In Figs 1F, S1 and S2, it would be helpful to use image analysis software to quantify the amount of nuclear cGAS across a large number of cells.
4. Please provide a supplementary figure outlining the gating strategy for Fig. 3.
5. Please provide protein gels showing purity of the proteins used in the assay shown in Fig. 6.

Minor points

6. In the title, please replace "hence" with "and".
7. cGAS is the abbreviation for "cGAMP synthase" or "cyclic GMP-AMP synthase", not "cyclic cGMP-AMP synthase". Please correct abstract and introduction.
8. On page 7, please refer to Fig S9E and S9F (not S10E and S10F)
9. Could the authors provide a reference for BMDMs? Using cells from BMDM cultures at an early timepoint seems unusual. What is the phenotype of these cells?
10. Fig S9 should be moved into the main manuscript to form Fig 7.
11. Please cite and briefly discuss PMID 28279982, which shows that cGAS-STING induce ISGs after IR.
12. Please cite and briefly discuss PMID 30827685, which shows that cGAS is localised at the plasma membrane and suggests that nuclear localization may be an artefact of cell lysate preparation.

1st Revision - authors' response

5th August 2019

Point by point responses to reviewers

We are grateful to this reviewer for the positive comments and constructive suggestions on our manuscript. We have now included the additional data and made all the suggested changes. We feel that their comment have improved the manuscript and hope that it will now be published swiftly in the EMBO Journal. Please see below our detailed response (in plain text) to the specific reviewer comments (highlighted in bold)

Referee #1:

Specific experimental points:

Point #a:

The authors perform biochemical assays to reveal the mechanism behind HR-mediated DNA repair inhibition using the cGAS catalytic domain (Figure 6). In contrast, cell assays are based on full-length cGAS. To have these two sets of data comparable, it would be important to see if the cGAS catalytic domain is sufficient to inhibit DNA repair in cells as well.

Response: We have performed this experiment. The cGAS catalytic domain also inhibits the HR-DNA repair (**Figure EV5B**). Please note that in addition to the cell

based assay (**Figure EV4A, B**), we also provide evidence that the purified full length cGAS and catalytic domain cGAS do inhibit D-loop formation (**Figure 8B, C and I**).

Points #b & c:

b) Specific blocking of D-loop formation by cGAS in vitro (Fig. S8). I find the conclusions of these experiments a bit premature. First, the conclusions of the authors that cGAS and MHF have similar binding affinities is not entirely justified, because the presented data may just well reflect a titration of binding sites (thus underestimating the affinity). In fact, the curve in Fig. S8 looks a bit like a linear slope immediately followed by a plateau which is characteristic for a titration experiment of a tight binder. If the authors want to make that conclusion, perhaps they could use a more quantitative, equilibrium assay such as fluorescence polarization anisotropy?

c) Specific blocking of D-loop formation by cGAS in vitro (Fig. S8). Along the same lines. I find it rather surprising that MHF does not block D-loop formation at all at a concentration where it robustly binds DNA. Shouldn't there be at least a reduction simply due to competition with Rad51 for the dsDNA? Perhaps the conclusion is the other way around, i.e. MHF specifically allows D-Loop formation even though it is bound by DNA, while cGAS is a mere steric competitor? To draw the conclusion that cGAS "specifically" inhibits D-loop is not yet justified by these experiments.

Response to points b & c: We thank the reviewer for this point. The point of the MHF experiment was to test whether inhibition of D-loop by cGAS was simply due to its DNA binding, and if so, whether other DNA binding proteins could inhibit D-loop formation. The MHF experiments suggested that DNA binding alone was not sufficient for D-loop inhibition. This prompted us to analyze cGAS mutants for features essential for HR inhibition. We found that inhibition RAD51-mediated strand invasion by cGAS is due to its ability to self-oligomerize thereby condensing bound template dsDNA to a higher-order state less amenable to invasion by RAD51 filaments. Therefore inhibition of RAD51-mediated strand invasion by cGAS and not MHF could be explained by inherent differences in their ability to condense bound DNA via self-oligomerization.

Suggested Editorial changes:

- The introduction is quite brief and more background on the topics concerning this study would be highly beneficial to the general readership.

Response: We have expanded the instruction as suggested.

- Also, rationales behind experiments are sometimes lacking, for instance Figure 1a-c. What is the rational of studying low and high density cultures and +/- serum?

Response: We have added more text to better explain the rationale for the experiments. The aim of the low and high density cultures in **Figure 1** was to study the effect of arresting cells at G0/G1 by contact inhibition at high density and to compare it with cycling cells at low density cultures (mainly at S/G2).

- "DNA binding zinc finger" C396A/C397A mutant: please rephrase since the effect of the mutant is loss of DNA binding and it is a bit confusing as written.

Response: We have rephrased as suggested.

- P6 "due to the recruitment of these factor to chromatin upon DNA damage". These "factors"?

- Fig. S8E typo: "Proetin"

Response: Thank you for spotting these typos. We have carefully gone through the text and corrected the typos.

Referee #2:

Major points

Point #1: Are cGAS-deficient cells also more resistant to other genotoxic stresses, apart from IR? Such data would broaden the study.

Response: Yes. cGAS Knockout (KO) cells are more resistant to other genotoxic stressor for example etoposide and doxorubicin. These data are part of an ongoing work on a separate project on the impact of nuclear cGAS in anti-tumor therapy. Therefore we request not to include data in the present manuscript primarily focused on describing the role and mechanisms of cGAS in DNA repair.

Point #2. The authors suggest that cGAS is bound to chromatin "constantly" (Fig 5). Most of the evidence for this comes from experiments with overexpressed, GFP-tagged cGAS. Have the authors estimated how much endogenous cGAS protein molecules a cell contains, relative to the size of its DNA genome? Is cGAS essentially behaving like histones, binding the whole genome? It seems hard to imagine there is enough cGAS protein expressed. Alternatively, does cGAS bind specific regions in the genome, as suggested in PMID 30811988? At a minimum, this reference and PMIDs 30270045 & 28738408, which report assays on chromatin binding by cGAS, should be discussed.

Response: We have demonstrated that endogenous cGAS is constantly present in the nucleus as chromatin-bound protein in more than 5 different cell types including BMDMs (Figure 1, Figure 2E, Figure 3C, Figure 6C, D), BMDMs, HeLa cells, Raw 264.7 macrophages, THP1 cells (Figure EV1D). We have analysed the abundance of chromatin-bound endogenous cGAS in BMDMs. It is very abundant (data not shown). However, here it is worth to emphasize that inhibition of RAD51-mediated strand invasion by cGAS is not simply via its binding to DNA but due to its ability to self-oligomerize and therefore condense dsDNA into a higher-order state. Therefore cGAS does not have to coat all possible DNA binding sites in the genome for this inhibition to occur. As discussed, our proposed mechanisms is perhaps analogous to that by proteins like Histone H1 that restrain HR-DNA repair by promoting chromatin compaction (e.g. PMID: 12820979, PMID: 24798879, PMID: 17613284). Similarly, as pointed out by the reviewer, we don't imagine that inhibition of HR-DNA repair such proteins requires them occupying all possible DNA binding sites in the genome.

Point #3: In Figs 1F, S1 and S2, it would be helpful to use image analysis software to quantify the amount of nuclear cGAS across a large number of cells.

Response: We have performed the quantification of nuclear cGAS as suggested. Please see Figure 1A-C, and Figure EV2B).

Point #4: Please provide a supplementary figure outlining the gating strategy for Fig. 3.

Response: We have included the gating strategy as suggested. Please see Appendix Fig S3.

Point #5: Please provide protein gels showing purity of the proteins used in the assay shown in Fig. 6.

Response: We have included the protein gels showing purity of cGAS and component used in D-loop assay. Please see Figure EV5A.

Minor points

6. In the title, please replace "hence" with "and".

Response: We have changed the title as suggested.

7. cGAS is the abbreviation for "cGAMP synthase" or "cyclic GMP-AMP synthase", not "cyclic cGMP-AMP synthase". Please correct abstract and introduction.

Response: We have made the correction as suggested.

8. On page 7, please refer to Fig S9E and S9F (not S10E and S10F)

Response: We have carefully gone through the manuscript and ensured that the Figures are correctly called out.

9. Could the authors provide a reference for BMDMos? Using cells from BMDM cultures at an early timepoint seems unusual. What is the phenotype of these cells?

Response: The standard protocols for generating bone marrow derived macrophages (BMDMs) is to culture bone marrow cells in 20% L929 conditioned medium for 10 days. To generate bone marrow differentiating monocytes (BMDMos), we culture bone marrow cells in 20% L929 conditioned medium for 4-5 days. In contrast to the terminally differentiated macrophages (BMDMs), BMDMos are cycling cells with monocyte phenotype. We do not have a specific reference but this is the protocol that established and have successfully been using in our lab for many years. We have described this in the manuscript

10. Fig S9 should be moved into the main manuscript to form Fig 7.

Response: We have done as suggested.

11. Please cite and briefly discuss PMID 28279982, which shows that cGAS-STING induce ISGs after IR.

Response: We have cited this work.

12. Please cite and briefly discuss PMID 30827685, which shows that cGAS is localised at the plasma membrane and suggests that nuclear localization may be an artefact of cell lysate preparation.

Response: We have cited this study.

2nd Editorial Decision

12th August 2019

Thank you for submitting your revised manuscript for our consideration. I have now gone through the revised manuscript and assessed your responses to the referee reports, and I am pleased to see that all scientific issues have been satisfactorily addressed. We should therefore be ready to accept the study for publication in The EMBO Journal, pending several remaining formal/editorial modifications.

2nd Revision - authors' response

19th August 2019

Thank you for the offer to accept our work for publication in the EMBO Journal. We have now made the suggested editorial changes which you will find highlighted in correction mode.

Accepted

2nd September 2019

Thank you for submitting your final revised manuscript for our consideration. I am pleased to inform you that we have now accepted it for publication in The EMBO Journal.

Corresponding Author Name: Nelson Ongondo Gekara

Manuscript Number: EMBOJ-2019-102718